# Vinculin–Arp2/3 interaction inhibits branched actin assembly to control migration and proliferation

John James[1], Artem I Fokin[1,2], Dmitry Y Guschin[1], Hong Wang[3], Anna Polesskaya[1], Svetlana N Rubtsova[1], Christophe Le Clainche[3], Pascal Silberzan[2], Alexis M Gautreau[1], Stéphane Romero[1]

**Vinculin is a mechanotransducer that reinforces links between cell adhesions and linear arrays of actin filaments upon myosin-mediated contractility. Both adhesions to the substratum and neighboring cells, however, are initiated within membrane protrusions that originate from Arp2/3-nucleated branched actin networks. Vinculin has been reported to interact with the Arp2/3 complex, but the role of this interaction remains poorly understood. Here, we compared the phenotypes of vinculin knock-out (KO) cells with those of knock-in (KI-P878A) cells, where the point mutation P878A that impairs the Arp2/3 interaction is introduced in the two vinculin alleles of MCF10A mammary epithelial cells. The interaction of vinculin with Arp2/3 inhibits actin polymerization at membrane protrusions and decreases migration persistence of single cells. In cell monolayers, vinculin recruits Arp2/3 and the vinculin–Arp2/3 interaction participates in cell–cell junction plasticity. Through this interaction, vinculin controls the decision to enter a new cell cycle as a function of cell density.**

## Introduction

Actin dynamics control cell shape, adhesion, and migration (Pollard & Cooper, 2009). Actin filaments form linear and branched arrays (Le Clainche & Carlier, 2008). Whereas branched actin arrays generate pushing forces through the Arp2/3 complex, linear arrays can generate pulling forces through myosin-mediated contractility (Pollard, 2016; Garrido-Casado et al, 2021). Indeed, cell migration requires a combination of forces to drive protrusion of the plasma membrane and to pull on the substratum (Blanchoin et al, 2014).

The RAC1-WAVE-Arp2/3 pathway drives membrane protrusions (Papalazarou & Machesky, 2021; Bieling & Rottner, 2023). This pathway is embedded in positive feedback loops that sustain the membrane protrusion at the front edge of a migrating cell and increase the persistence of migration (Krause & Gautreau, 2014). Signaling from branched actin at the cell cortex is also critical for cells to progress into the cell cycle by impinging on tumor suppressor genes controlling the G1/S transition (Molinie et al, 2019). Various Arp2/3 complexes coexist in the same cells through the combinatorial assembly of paralogous subunits (Pizarro-Cerdá et al, 2016). For example, ARPC1B-containing Arp2/3 complexes are more efficient at nucleating branched actin and at forming stable branched junctions than ARPC1A-containing Arp2/3 complexes (Abella et al, 2016). ARPC1B-containing Arp2/3 complexes, but not the ones containing ARPC1A, generate cortical branched actin that drives persistent migration and delivers a signal for cell cycle progression (Molinie et al, 2019).

Vinculin is recruited by mechanosensitive proteins that sense forces exerted in cell adhesions, and responds to these forces by reinforcing the linear arrays of actin filaments attached to cell adhesions (Bays & DeMali, 2017). Vinculin is composed of a head that interacts with adhesion proteins and a tail that binds to actin filaments (Humphries et al, 2007; Le Clainche et al, 2010). This ability of vinculin to link actin filaments to adhesion sites is inhibited by an intramolecular interaction masking relevant binding sites of the head and the tail (Atherton et al, 2016). In adhesions to the extracellular matrix (ECM), the vinculin head is recruited to cryptic binding sites in talin that are exposed upon stretching because of myosin-mediated contractility (del Rio et al, 2009; Ciobanasu et al, 2014; Yao et al, 2014a). The vinculin head is similarly recruited to a cryptic binding site of α-catenin at cell–cell adhesions upon myosin-mediated contractility of actin filaments (Yao et al, 2014b; Seddiki et al, 2018; Vigouroux et al, 2020). Force-dependent recruitment of vinculin results in its activation allowing the vinculin tail to link additional actin filaments and thereby reinforce the cytoskeletal attachment of cell adhesions. In the process, vinculin mechanotransduces signals, because vinculin-depleted cells exhibit enhanced proliferation, survival, and anchorage-independent growth (Fernández et al, 1993; Subauste et al, 2004; DeWane et al, 2023).

[1]Laboratory of Structural Biology of the Cell (BIOC), CNRS UMR7654, École Polytechnique, Institut Polytechnique de Paris, Palaiseau, France   [2]Laboratoire PhysicoChimie Curie UMR168, Institut Curie, Paris Sciences et Lettres, Centre National de la Recherche Scientifique, Sorbonne Université, Paris, France   [3]Université Paris-Saclay, CEA, CNRS, Institute for Integrative Biology of the Cell (I2BC), Gif-sur-Yvette, France

Correspondence: alexis.gautreau@polytechnique.fr; stephane.romero@polytechnique.edu
Dmitry Y Guschin's present address is Scientific center for Translational Medicine, Sirius university of Science and Technology, Sirius, Russia
Svetlana N Rubtsova's present address is Institute of Carcinogenesis, N.N. Blokhin National Medical Research Center of Oncology, Moscow, Russia

The cytoskeleton reinforcement function of vinculin is in line with the fact that cell adhesions mature over time. Adhesions to the ECM mature from nascent adhesions at the edge of membrane protrusions into focal adhesions (FAs) as the leading edge moves forward and myosin motors exert contractility on newly formed adhesions (DePasquale & Izzard, 1991; Alexandrova et al, 2008; Thievessen et al, 2013). Similarly, E-cadherin–based adherens junctions (AJs) form initial interdigitations that mature into straight cell–cell adhesions, as contractility develops (Kovacs et al, 2002; Leerberg et al, 2014; Li et al, 2020, 2021). Vinculin was shown in vitro to remodel branched actin networks into bundles (Boujemaa-Paterski et al, 2020). In this context, the reported interaction of vinculin with Arp2/3, the major player in protrusion formation, is particularly intriguing.

Vinculin binds to the Arp2/3 complex through the linker that connects its head to its tail (DeMali et al, 2002). This linker is not masked by the head-to-tail intramolecular interaction (Borgon et al, 2004), suggesting that the interaction should be independent from vinculin activation. Moreover, this interaction is likely to occur directly as shown by an in vitro GST pulldown (DeMali et al, 2002). Subsequently, "vinculin–Arp2/3 hybrid complexes" were purified from chicken smooth muscles (Chorev et al, 2014). These hybrid complexes lack some subunits of the canonical Arp2/3, namely, ARPC1, ARPC4, and ARPC5 subunits, and do not involve additional proteins, confirming a direct interaction. These observations suggest the existence of constitutive vinculin–Arp2/3 complexes. However, this interaction is regulated by EGF stimulation, RAC1 activity, and Src-dependent phosphorylations of vinculin on tyrosine residues that are located far away from the Arp2/3 binding site, but that contribute to vinculin activation (DeMali et al, 2002; Zhang et al, 2004; Moese et al, 2007; Auernheimer et al, 2015). The point mutation P878A in vinculin was shown to impair its interaction with Arp2/3, and a re-expression of this mutant in vinculin-null MEFs does not rescue defective protrusions and spreading of these cells, unlike WT vinculin (DeMali et al, 2002). Together, these data suggest that vinculin could activate Arp2/3 and form membrane protrusions. However, the biochemical effects of vinculin on Arp2/3 remain largely unexplored (Romero et al, 2020).

Here, we used epithelial cells to investigate the role of the vinculin–Arp2/3 interaction. We were able to distinguish the specific subset of functions that the vinculin–Arp2/3 interaction endows, compared with the more general role of vinculin in the cytoskeletal reinforcement of cell adhesions. We found that the vinculin–Arp2/3 interaction inhibits actin assembly in membrane protrusions, and therefore antagonizes branched actin in the control of cell migration and cell cycle progression, and regulates at cell–cell junction plasticity through Arp2/3 recruitment.

## Results

### The vinculin linker enhances membrane protrusion and migration persistence through Arp2/3 binding

We first examined the interaction of vinculin with the Arp2/3 complex in the MCF10A cell line, where we have characterized

the role of the Arp2/3 complex in cell migration (Molinie et al, 2019; Simanov et al, 2021). MCF10A cells are human mammary epithelial cells, which are immortalized but not transformed (Soule et al, 1990). Importantly, this cell line is diploid for the most part of its genome (Worsham et al, 2006; Kadota et al, 2010). We found that the exogenous expression of tagged full-length vinculin was down-regulated over time by MCF10A cells, although these cells stably transfected retained the antibiotic resistance. This did not occur for the exogenous expression of a construct limited to the vinculin linker that connects the head to the tail and that contains the Arp2/3 binding site. We obtained stable MCF10A cell lines expressing the GFP-tagged linker of vinculin (amino acids 811–881) or its derivative containing the point mutation P878A that impairs Arp2/3 binding (DeMali et al, 2002). Importantly, expression levels of endogenous vinculin and Arp2/3 subunits were not modified in the two cell lines (Fig 1A). When mutated, the linker indeed bound much less efficiently to the Arp2/3 complex despite an expression level similar to the WT linker (Fig 1B). Interestingly, the immunoprecipitate of the vinculin linker contained the Arp2/3 subunit ARPC1B, which is not present in the vinculin–Arp2/3 hybrid complexes that were purified from tissues (Chorev et al, 2014), suggesting that in our cell system, the vinculin linker binds to the canonical Arp2/3 complex.

MCF10A cells expressing the WT linker, but not its P878A derivative, exhibited extensive membrane protrusions at their periphery (Fig 1C, Video 1). Cells expressing the vinculin linker were significantly more spread (Figs 1D and S1A) and migrated more persistently than parental cells (Figs 1E and F and S1B), two readouts of cortical polymerization of branched actin. Decreased cell speed and mean square displacement (MSD) were associated with this increased persistence of cell migration (Fig S2). Decreased speed and MSD are often associated with increased cortical branched actin in MCF10A cells, for example, when RAC1 is activated by mutation or when the Arp2/3 inhibitory protein Arpin is down-regulated (Molinie et al, 2019). In contrast, cell migration persistence systematically correlates with increased polymerization of cortical branched actin (Dang et al, 2013; Molinie et al, 2019; Simanov et al, 2021).

We then examined membrane protrusions that power cell migration. The extension of membrane protrusions depends on the efficiency of the molecular clutch that couples cell adhesions to branched actin polymerizing against the plasma membrane (Thievessen et al, 2013; Hirata et al, 2014). Thus, the speed of membrane protrusions is a function of both the rate of actin assembly and actin rearward flow. Upon mCherry–actin expression, we were able to measure speed and the length of the protrusions, defined as the distance between the protrusion edge and the first transverse arc at the base of protrusions (Fig 1G). Protrusions were longer and protruded 1.7-fold faster when cells expressed the WT vinculin linker, but not the P878A derivative (Fig 1H–J). Fluorescent actin also allowed us to image the rearward flow of the cytoskeleton using total internal reflection fluorescence structured illumination microscopy (TIRF-SIM, Video 2). We found that both the rearward flow, relative to the substratum, and the calculated actin assembly rate (see the Materials and Methods section), relative to the leading edge, were significantly increased in cells expressing the vinculin linker (Fig 1I, K, and L). However, protrusion efficiency, the ratio of protrusion speed to actin assembly rate, was slightly decreased in

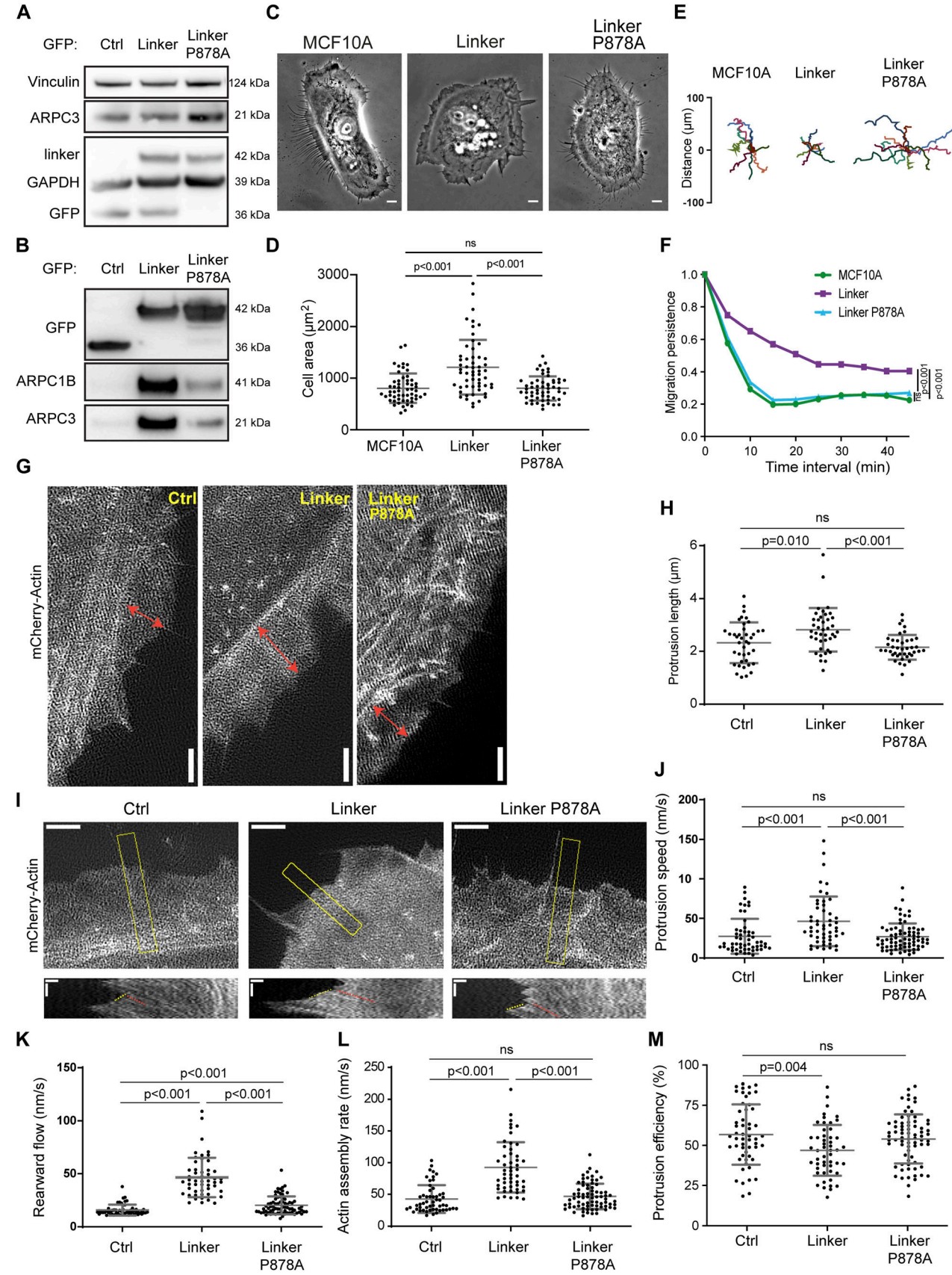

cells expressing the vinculin linker, indicating that protrusion speed does not fully scale with the actin assembly rate (Fig 1M). This effect on protrusion efficiency depended on the interaction between the vinculin linker and Arp2/3 because the expression of the P878A derivative does not modify the actin assembly rate, nor protrusion speed, compared with control cells (Fig 1I, K, L, and M). Together, these results suggest either that vinculin activates the Arp2/3 complex through the linker domain, or that vinculin inhibits Arp2/3 and the linker behaves as a dominant-negative fragment. Therefore, we produced and purified the vinculin linker in its WT and P878A forms. In vitro, these proteins did not modify the kinetics of spontaneous actin polymerization, nor Arp2/3-mediated branched nucleation (Fig S3), indicating that the observed effects of the vinculin linker in cells described above require additional factors or post-translational modifications that are only present in cells to regulate Arp2/3.

### Vinculin knock-in of the point mutation that impairs the Arp2/3 interaction increases actin dynamics and migration persistence as vinculin knock-out does

Because no activity of the linker can be revealed in vitro, we generated knock-outs of the *VCL* gene that encodes vinculin to distinguish between the linker-mediated activation and dominant-negative hypotheses. We transfected MCF10A cells with purified Cas9 and a guide RNA (gRNA) that targets the beginning of the ORF in the first exon of the *VCL* gene. Cas9-mediated double-strand breaks (DSBs) are frequently repaired by the error-prone non-homologous end-joining (NHEJ) mechanism (Jasin & Haber, 2016). This method using the purified Cas9 and gRNA was sufficiently efficient to identify KO clones by Western blot–based screening without selection (Fig 2A). The genomic DNA has been sequenced, and we identified two KO clones with different frameshifts within the two alleles of *VCL* (Fig 2B), for further characterization. Those mutations lead to the production of peptides that share only the three first N-terminal residues with vinculin (Fig S4A). As expected, focal adhesions (FAs) of these two clones were not stained by vinculin antibodies (Fig 2C and D). Paxillin-stained FAs were significantly more elongated, by 60% on average, in vinculin KO compared with parental cells (Fig 2E), but cell adhesion to a fibronectin-coated surface was not significantly modified (Fig S4B).

To specifically examine the role of the Arp2/3 interaction, we designed a strategy based on homology-directed repair to introduce the P878A point mutation in the endogenous *VCL* gene of MCF10A cells. We transfected MCF10A cells with three plasmids and a long repair oligonucleotide that provides the P878A mutation together with a PvuII restriction site. One of the plasmids provided Cas9, the second plasmid encoded a gRNA that allows Cas9 to cut the *VCL* gene close to the P878 codon, and the third one encoded a gRNA that targets the *ATP1A1* gene. *ATP1A1* encodes the ubiquitous and essential sodium–potassium ion pump, which is the target of the ouabain drug. The gRNA targets the ATP1A1 region that is directly recognized by ouabain. Upon ouabain treatment, only cells that have repaired the *ATP1A1* DSB by NHEJ so as to introduce an in-frame indel produce an ion pump that is both functional and insensitive to ouabain (Agudelo et al, 2017). The ouabain selection is more efficient than more classical antibiotic selection of transfected plasmids, because it ensures that Cas9-mediated DSBs are efficiently produced in the selected cells, and not only that the selected plasmid is present. The DSBs in *VCL* are often repaired by NHEJ, but can also be repaired by homology-directed repair using the provided oligonucleotide as a template. After extensive screening of clones by PvuII restriction of the PCR-amplified genomic region, we managed to isolate the desired knock-in (KI-P878A) clone, where the P878A mutation was introduced in both alleles of *VCL* (Fig 2F and G). The expression level of P878A vinculin was similar to WT vinculin (Fig 2H). P878A vinculin properly localized to FAs (Fig 2I and J) and did not impact elongation of FAs (Fig 2K). Moreover, cell adhesion to fibronectin remained similar to parental and KO cells (Fig S4B). Altogether, this suggests that P878A vinculin is functional and that Arp2/3 binding is not required for the FA-related functions of vinculin in MCF10A cells.

We then evaluated KO and KI-P878A clones for their ability to migrate and produce membrane protrusions. Like the MCF10A clone that expressed the vinculin linker, KO clones exhibited more persistent trajectories than parental cells (Figs 3A and B and S5A, Video 3). They were also 20% more spread than parental cells (Fig 3E). The KI-P878A clone that expresses the vinculin protein containing the P878A substitution displayed a similar, but even more pronounced, phenotype of increased persistence and spreading than the KO clones that expressed no vinculin (Figs 3C, D, and F and S5B and D). This suggests that the vinculin–Arp2/3 interaction is critical for these functions. We next investigated whether the vinculin–Arp2/3 interaction is involved in Arp2/3 recruitment at the leading edge of the lamellipodium. In KO clones, the intensity of Arp2/3 was not significantly different compared with parental cells, but slightly increased in KI-P878A cells (Fig 3G–I). However, the width of Arp2/3 staining in the lamellipodium was significantly increased in KO and KI-P878A cells compared with parental cells (Fig 3G, H, and J), indicating that the lack of vinculin–Arp2/3

**Figure 1. Expression of the vinculin linker that binds to the Arp2/3 complex increases actin polymerization and membrane protrusion.**
**(A, B)** Arp2/3 complex co-immunoprecipitates with the vinculin linker (amino acids 811-881). **(A, B)** MCF10A cells stably expressing GFP, the GFP-tagged linker in a WT or P878A form, were lysed and subjected either to Western blot analysis (A), or to GFP immunoprecipitation and Western blot analysis (B). N = 3; 1 representative experiment shown. **(C)** Phase-contrast images of the same cell lines. Scale bar: 5 μm. **(D)** Cell area of the cell lines stained for actin. Mean ± SD, n = 56, *t* test. N = 3; pooled measurements from the three independent repeats are plotted. **(E, F)** Single-cell migration. **(E, F)** Cell trajectories (E) and migration persistence (F). Mean ± SD, n = 35, linear mixed-effect model. N = 3; pooled measurements from the three independent repeats are plotted. **(G, H)** Membrane protrusions of the stable MCF10A cell lines expressing the vinculin linker transiently transfected with mCherry–actin. **(G, H)** TIRF-SIM images of mCherry–actin (G) and quantification of protrusion length (H). Double-headed arrows in red indicate the length of lamellipodia. Scale bar: 2 μm. Mean ± SD, n = 40, *t* test. N = 3; pooled measurements from the three independent repeats are plotted. **(I)** Kymographs (bottom panels, scale bars: 0.4 μm horizontal, 40 s vertical) were generated along a line centered in the region boxed in yellow in the TIRF-SIM video (scale bar: 2 μm). **(J, K)** Dashed yellow and red lines indicate protrusion speed (J) and rearward flow (K), respectively. **(L)** Actin assembly rate (L) is the sum of protrusion speed and rearward flow. **(M)** Protrusion efficiency (M) is the ratio of protrusion speed to actin assembly rate. Mean ± SD, n = 40, *t* test. N = 3 with similar results; pooled measurements from the three independent repeats are plotted.

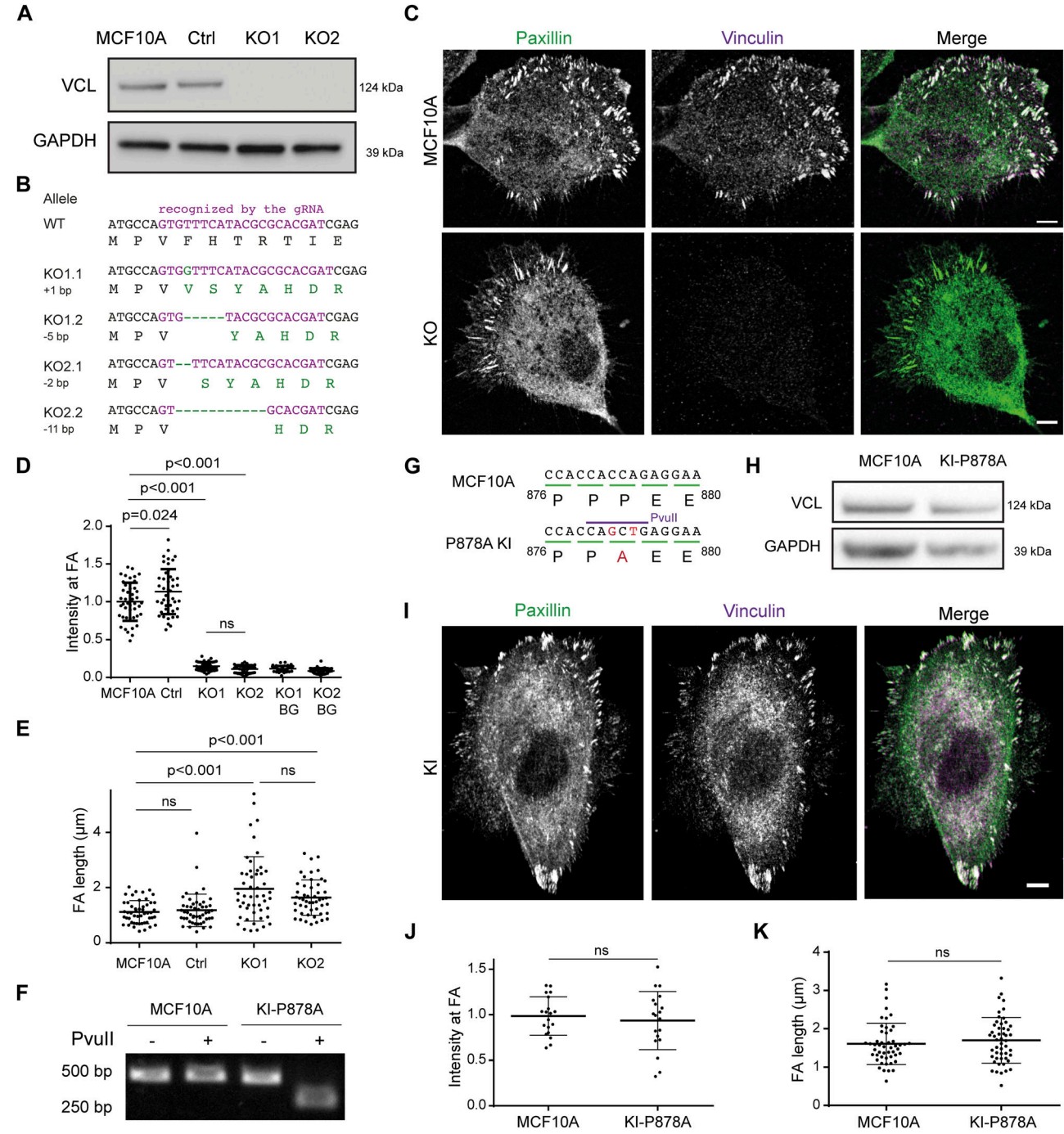

**Figure 2. Characterization of *VCL* knock-out and knock-in cell lines.**
**(A)** Parental MCF10A, and isolated clones transfected with a *VCL*-targeting gRNA or a non-targeting gRNA were analyzed by Western blot. **(B)** Sequences of the two alleles in each KO cell line. All mutations induce a frameshift and thus a premature stop codon in the ORF. **(C)** Staining of vinculin and paxillin in parental and KO cells. Scale bar: 5 μm. **(D)** Quantification of vinculin staining in focal adhesions (FAs) and normalization by the intensity of parental cells. BG refers to the background in the non-FA cytoplasm. Mean ± SD, n = 45, *t* test. N = 3 with similar results; pooled measurements from the three independent repeats are plotted. **(E)** Quantification of the length of FAs. Mean ± SD, n = 45, *t* test. N = 3 with similar results; pooled measurements from the three independent repeats are plotted. **(F)** Genome analysis of the KI. Part of the *VCL* ORF containing the P878A mutation was amplified by PCR and digested with the PvuII restriction enzyme. Agarose gel electrophoresis of digested or undigested PCR fragment. **(G)** Sequencing of the genome-amplified PCR fragment confirmed the presence of the P878A mutation on the two alleles and the introduction of the PvuII restriction site in the genome of the KI-P878 line. **(H)** Cell lysates of parental MCF10A, and the isolated clone of the KI-P878A were analyzed by Western blot using anti-vinculin and anti-GAPDH antibodies. **(I)** Staining of vinculin and paxillin in the KI-P878A cells. Scale bar: 5 μm. **(J)** Quantification of vinculin staining in focal adhesions (FAs) and normalization by the intensity of parental cells. Mean ± SD, n = 19, *t* test. N = 3 with similar results; pooled measurements from the three independent repeats are plotted. **(K)** Quantification of the length of FAs. Mean ± SD, n = 50, *t* test. N = 3 with similar results; pooled measurements from the three independent repeats are plotted. Ctrl refers to the MCF10A cell line that has been genome-edited using a Ctrl gRNA.

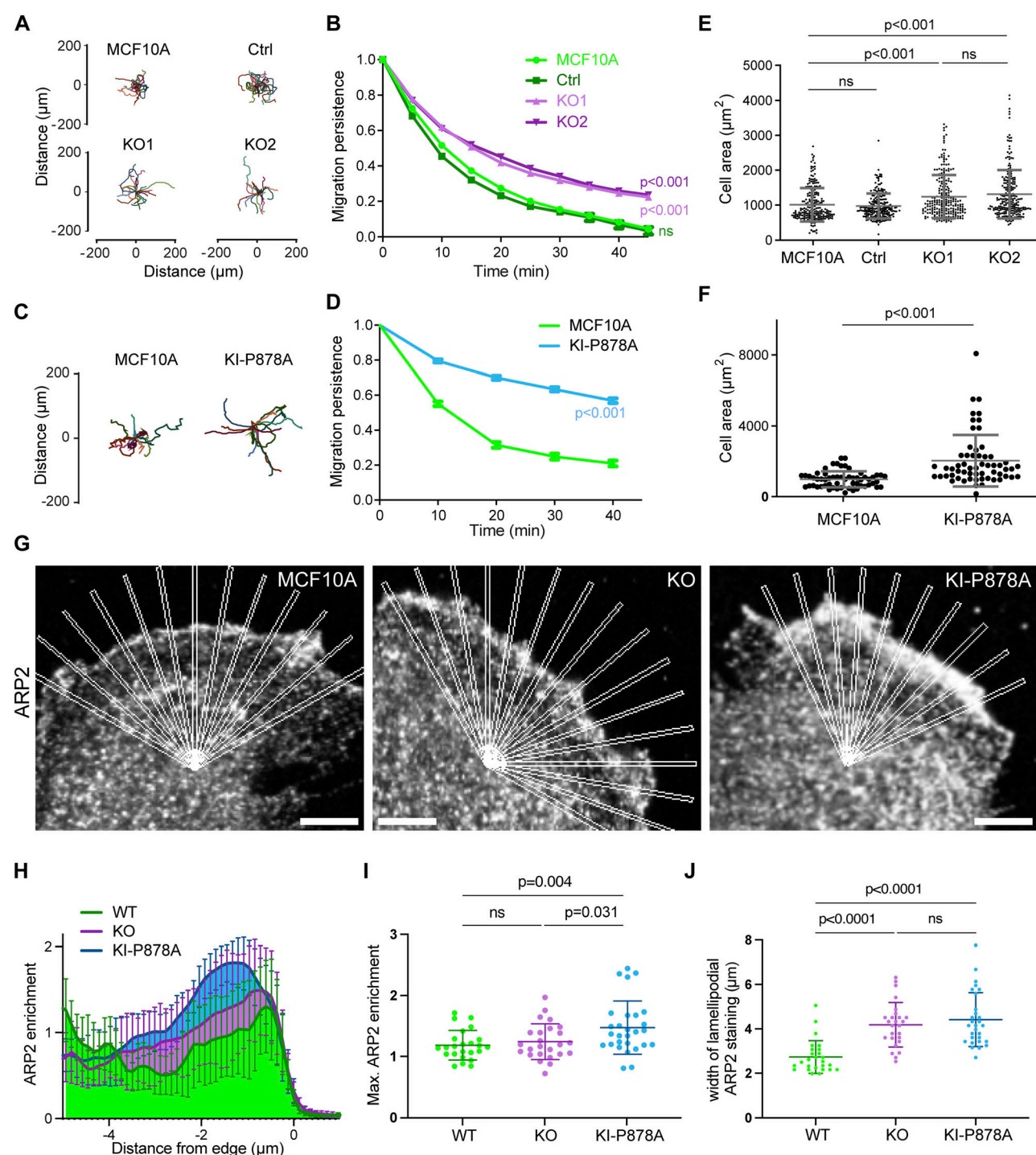

**Figure 3. Vinculin decreases migration persistence and Arp2/3 recruitment in membrane protrusions through its interaction with the Arp2/3 complex.**
**(A, B, C, D)** Single-cell migration of KO and KI-P878A cells. Cell trajectories (A, C) and migration persistence (B, D). **(B, D)** Mean ± SD, n = 75 in (B) and n = 35 in (D), linear mixed-effect model. N = 3; pooled measurements from the three independent repeats are plotted. **(E, F)** Cell area of KO and KI-P878A lines. **(E, F)** Mean ± SD, n = 228 in (E) and n = 59 in (F), t test. N = 3; pooled measurements from the three independent repeats are plotted. **(G)** Staining of ARP2 in the lamellipodium of parental, KO, and KI-P878A cells on single-plane confocal microscopy images. Scale bar: 5 $\mu m$. **(H)** Recruitment of ARP2 assessed by multiple radial line scans. Average profiles of ARP2 enrichment in the lamellipodium of parental, KO, and KI-P878A cells upon registering line scans to the cell edge. **(I, J)** Maximum intensity (I) and average width (J) of ARP2 staining at the edge of the lamellipodium. Mean ± SD, n = 29, t test. N = 3 with similar results; pooled measurements from the three independent repeats are plotted.

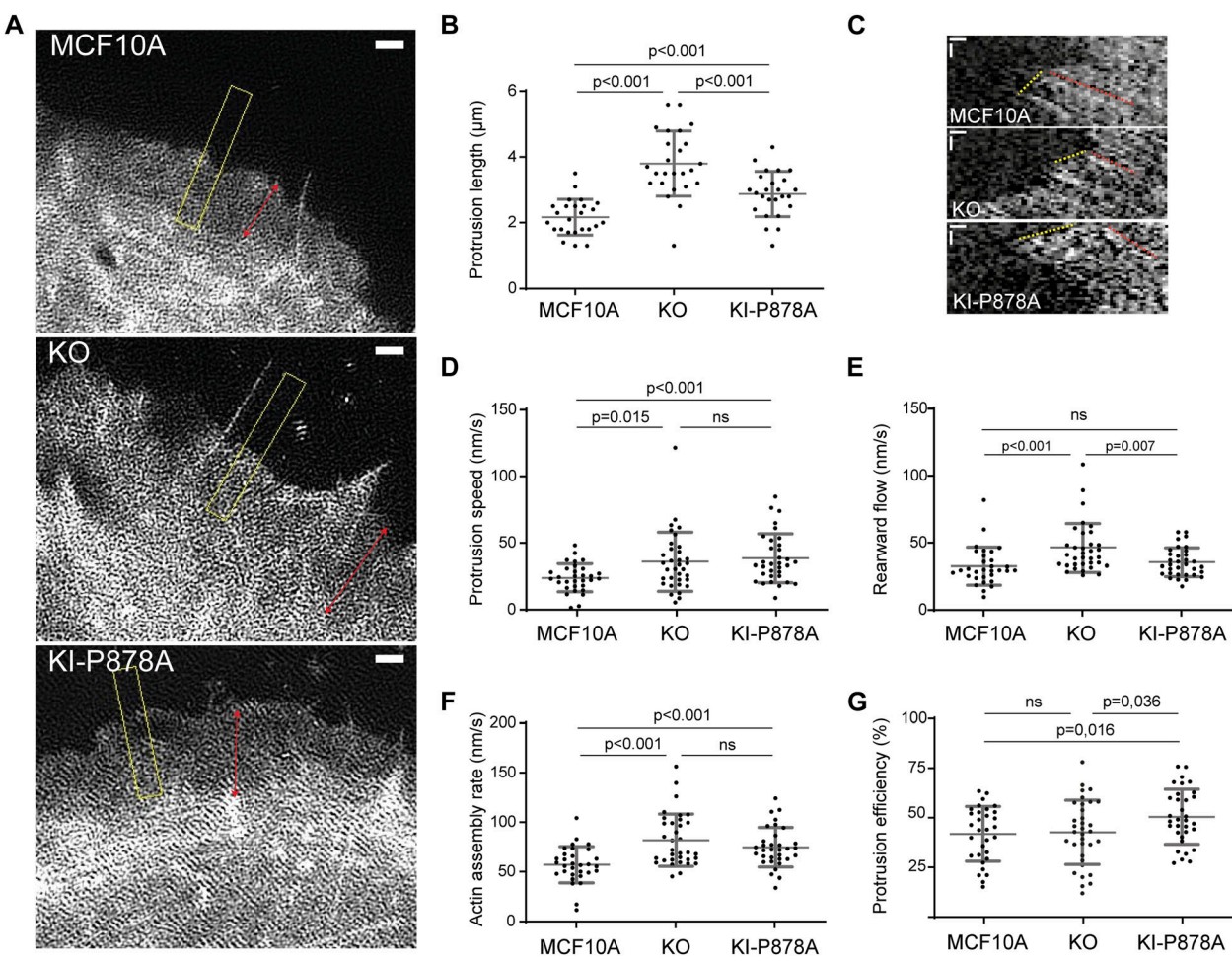

**Figure 4. Vinculin–Arp2/3 interaction decreases actin dynamics in membrane protrusions.**
**(A, B)** Membrane protrusions of the parental MCF10A, KO, and KI-P878A cell lines transiently transfected with mCherry–actin. **(A, B)** TIRF-SIM images of mCherry–actin (A) and quantification of the protrusion length (B). Double-headed arrows in red indicate the length of lamellipodia, and the dashed yellow box indicates the position of kymograph analysis. **(A)** Double-headed red arrows measure lamellipodium length (A). Scale bar: 1 μm. Mean ± SD, n = 25, *t* test. N = 3 with similar results; pooled measurements from the three independent repeats are plotted. **(A, C, D, E, F, G)** Kymograph analysis of KO and KI-P878A lines drawn along a line centered in the region boxed in yellow depicted in panel (A). **(C)** Scale bars: 0.25 μm horizontal, 10 s vertical (C). **(D, E)** Protrusion speed (D) is measured from yellow dashed lines, and rearward flow (E), from red dashed lines. **(F)** Actin assembly rate (F) is the sum of protrusion speed and rearward flow. **(G)** Protrusion efficiency (G) is the ratio of protrusion speed to actin assembly rate. Mean ± SD, n = 29, *t* test. N = 3 with similar results; pooled measurements from the three independent repeats are plotted. Ctrl refers to the MCF10A cell line that has been genome-edited using a Ctrl gRNA.

interaction results in a higher amount of branched actin in the lamellipodium.

We then examined actin dynamics in membrane protrusions by transiently transfecting our KO and KI-P878A clones with mCherry–actin. In agreement with Arp2/3 staining, the length of membrane protrusions was increased in both KO and KI-P878A clones compared with parental cells (Fig 4A and B). The increase was, however, significantly higher in KO than in KI-P878A cells for this parameter. In KO cells, the actin rearward flow was increased, indicative of a less efficient molecular clutch in the absence of vinculin (Fig 4C, E, I, and K). Protrusions extended 1.5-fold faster in KO cells, than in parental cells, in line with an increase of the same order of the actin assembly rate (Fig 4C, D, and F, Video 4; see the Materials and Methods section for the calculation of the actin assembly rate). As a consequence, protrusion efficiency was not different in KO and parental cells (Fig 4G). In KI-P878A cells, protrusions extended and actin assembled as fast as in KO cells (Fig 4C, D, and F, Video 4). But because the actin rearward flow measured in KI-P878A cells was similar to that of parental cells, the clutch was not deficient in this case (Fig 4E), resulting in protrusions of KI-P878A cells 1.2-fold more efficient than protrusions of KO and parental cells (Fig 4G). Together, these results show that the vinculin–Arp2/3 interaction inhibits actin assembly at the leading edge of the lamellipodium, whereas the molecular clutch, which couples branched actin to cell adhesions, depends on the presence of vinculin, but not on its interaction with Arp2/3. Because phenotypes induced by the expression of the vinculin linker are most similar to vinculin KO phenotypes, we can conclude that the linker is dominant-negative over the functions of vinculin in the cell.

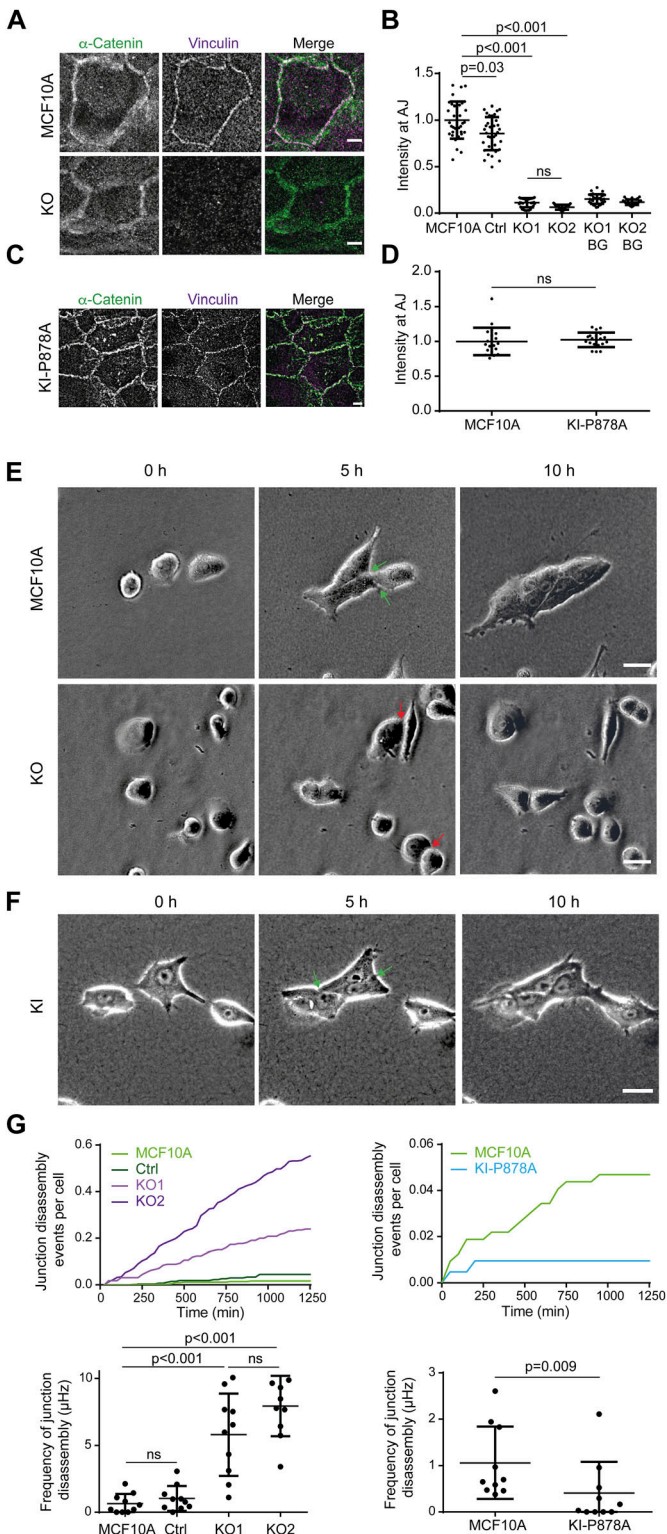

## Vinculin controls stability of cell–cell junctions and efficiency of collective migration

Because vinculin plays a key role in stabilizing cell–cell junctions (Bays & DeMali, 2017), we used our cell system to examine the potential role of the vinculin–Arp2/3 interaction in junction stability. We verified that KO cells displayed no vinculin staining at cell–cell junctions (Fig 5A and B). P878A vinculin expressed in KI-P878A cells was recruited at α-catenin–positive junctions as efficiently as the WT protein in parental cells (Fig 5C and D). Because on rigid 2D substrates such as petri dishes or glass coverslips, cell–cell interactions were not obviously affected in KO cells, we decided to embed cells into collagen gels to examine cell–cell interactions in a more physiological setting. In 3D collagen gels, behaviors of KO cells and parental cells were markedly different. Parental cells rarely dissociated when they met and formed multicellular slugs (Fig 5E, Video 5). In contrast, interactions between KO cells appeared not to engage them in a multicellular behavior (Fig 5E). KI-P878A cells did not present the asocial behavior of KO cells and formed multicellular "slugs" like parental cells (Fig 5F, Video 6). From videos, we were able to count the number of events, where a cell disengaged from cell–cell interactions after having met another cell or dissociating from slugs, per unit of time. The counts were then converted into frequencies expressed in $s^{-1}$ or Hertz. In this 3D setting, the frequency of junction disassembly was about 10-fold higher in KO than parental cells (Fig 5G). We verified that E-cadherin expression was not down-regulated in KO cells and that E-cadherin was properly recruited at cell–cell junctions in KO cells (Fig S6). Vinculin thus regulates the stability of junctions, but not E-cadherin–dependent cell–cell adhesion. In contrast, in KI-P878A cells, the quantification revealed that cell–cell junctions were significantly more stable than those of parental cells, dissociating 2.6-fold less (Fig 5G). Thus, the vinculin–Arp2/3-independent functions of vinculin are indeed critical to establish stable cell–cell junctions. Because the interaction of vinculin with Arp2/3 rather weakens cell–cell junctions, this interaction could play a role in cell–cell junction plasticity, to remodel junctions once stably formed.

We then analyzed collective migration of MCF10A, KO, and KI-P878A cells in a wound healing assay obtained by lifting an insert that initially constrains the monolayer (Poujade et al, 2007). The front edge of the monolayer of KO cells progressed significantly faster than parental cells and therefore covered the cell-free area more rapidly (Fig 6A and B, Video 7). Of note, more single cells were found at the front edge for KO than parental cells, reminiscent of less stable cell–cell junctions in KO cells. KI-P878A cells, although slower than KO cells, were also significantly faster than parental cells. Images were then analyzed by particle image velocimetry (PIV) to associate a displacement vector with each (x,y,t) coordinate (Petitjean et al, 2010). Cell speed was not only increased at the leading edge of the monolayer in KO and KI-P878A cells, but also

**Figure 5. Vinculin stabilizes cell–cell junctions.**
**(A, B, C, D)** Presence of vinculin at cell–cell junctions in KO and KI-P878A cells. **(A, B, C, D)** Staining of vinculin and α-catenin (A, C) and quantification of intensity at cell–cell junctions (B, D). Max z-projection from confocal microscopy. Scale bar: 5 μm. **(B, D)** Mean ± SD, n = 37 for (B) and n = 20 for (D), *t* test. N = 3 with similar results; pooled measurements from three independent repeats are plotted. **(E, F)** Time-lapse imaging of KO (E) and KI-P878A (F) cells in 3D collagen gels by phase contrast. Green arrows point at cell–cell junctions that were present at 5 h and that did not disassemble at 10 h, and red arrows point at cell–cell junctions that were present at 5 h and that were disassembled at 10 h. **(A, B)** Scale bars: 50 μm in (A); 25 μm in (B). **(G)** Quantification of cell–cell junction disassembly events per cell and frequency of cell–cell junction disassembly. Mean ± SD, n = 10, *t* test. N = 3 with similar results; pooled measurements from three independent repeats are plotted. Ctrl refers to the MCF10A cell line that has been genome-edited using a Ctrl gRNA.

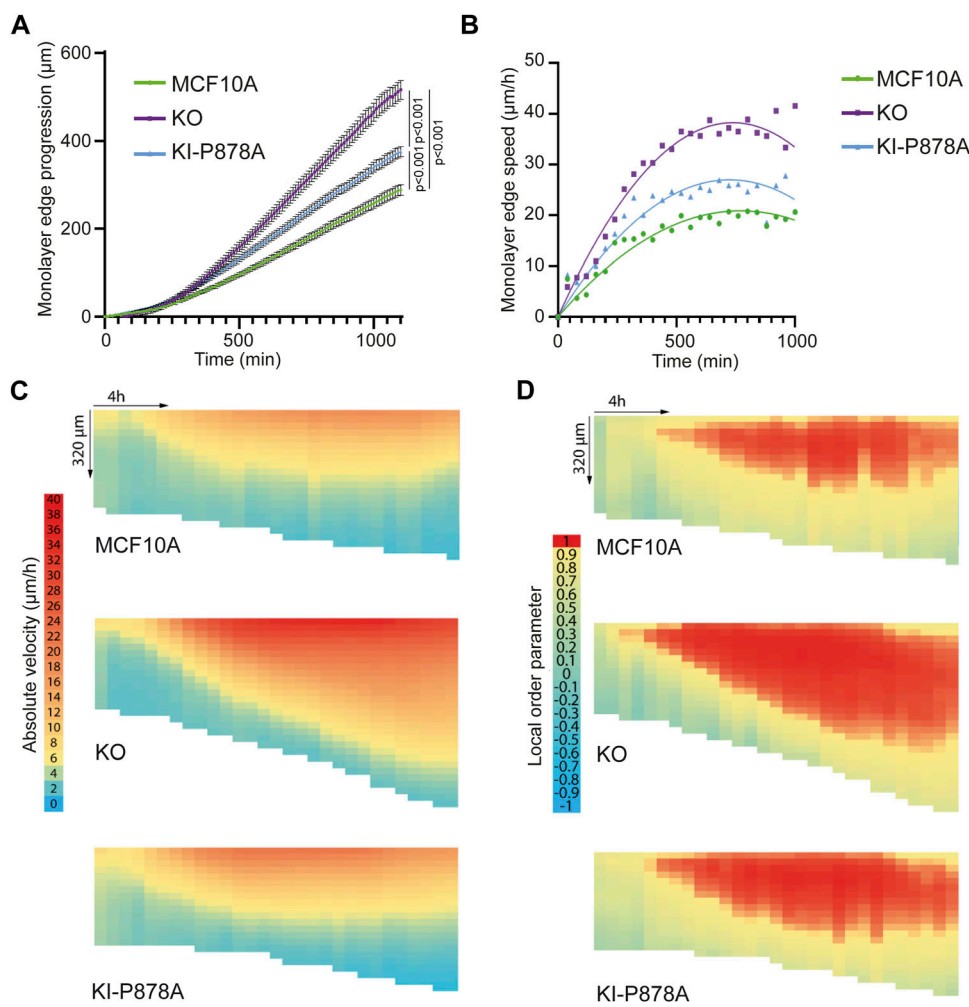

**Figure 6. Vinculin controls collective migration upon wound healing.**
Collective migration of MCF10A, KO, and KI-P878A cells over the wound was imaged by phase contrast over time and analyzed by particle image velocimetry. **(A)** Quantification of monolayer edge progression over time. **(B)** Quantification of monolayer edge speed over time. **(C, D)** Heat map representation of velocity ((C), length of displacement vectors) and local order parameter ((D), cosine of angles between adjacent displacement vectors). The vertical axis corresponds to coordinates along the perpendicular axis relative to the monolayer edge (edge position kept constant on the top of heat maps), whereas the horizontal axis corresponds to time course (from left to right). N = 3 with similar results; 1 representative experiment shown.

further transmitted backward, away from the edge, when compared to parental cells (Fig 6C). The local order parameter, which reflects the local alignment of displacement vectors, was also transmitted further backward in KO and KI-P878A cells when compared to parental cells (Fig 6D). However, the increase in both the cell speed and the local order parameter was further transmitted backward in KO cells, compared with KI-P878A cells, in line with the faster progression of the KO front edge. Thus, unlike single-cell migration, collective migration depends mainly on the Arp2/3-independent functions of vinculin, and partly on its Arp2/3 interaction.

### Vinculin recruits Arp2/3 at adherens junctions and restricts cell cycle progression through Arp2/3 binding

We then analyzed Arp2/3 recruitment by immunofluorescence at E-cadherin–positive AJs of KO and KI-P878A cells. In parental MCF10A cells, Arp2/3 was enriched at AJs, 6 h after plating; the amount of junctional Arp2/3 reached a maximum 1 d after plating and declined to a residual amount after 3 d (Fig 7A). Vinculin was present at AJs throughout junction maturation, even if it also peaked 1 d after plating like Arp2/3 (Fig S7). KO cells displayed reduced Arp2/3 staining at AJs compared with parental cells (Fig

7B). This was especially striking 1 d after cell plating, when no junctional Arp2/3 was detected. 6 h after plating, a low amount of Arp2/3 was detected at AJs of KO cells, but KO cells still recruited significantly less Arp2/3 than parental cells (Fig 7B). In KI-P878A cells, Arp2/3 recruitment at AJs was increased 6 h after plating compared with parental cells, and even more so compared with KO cells (Fig 7B and C). Yet, 1 d after plating, Arp2/3 recruitment at AJs of KI-P878A cells was dramatically reduced compared with parental cells, as in KO cells. Three days after plating, Arp2/3 at AJs is minimum in all three types of cells.

This kinetic analysis thus reveals two distinct phases of Arp2/3 recruitment at AJs. The early recruitment, at 6 h, only partially depends on the presence of vinculin and on its ability to interact with the Arp2/3. The later recruitment, after 1 d, fully depends on the presence of vinculin and on its ability to interact with the Arp2/3. Therefore, vinculin appears to retain Arp2/3 at AJs after an initial recruitment of Arp2/3 that does not depend on vinculin–Arp2/3 interactions. Stable MCF10A cell lines expressing GFP fusions with Arp2/3 subunits that are not part of the vinculin–Arp2/3 hybrid complexes revealed a GFP signal enriched at cell–cell junctions, 1 d after cell plating (Fig S8), thus confirming that vinculin interacts with the canonical Arp2/3 complex in MCF10A cells.

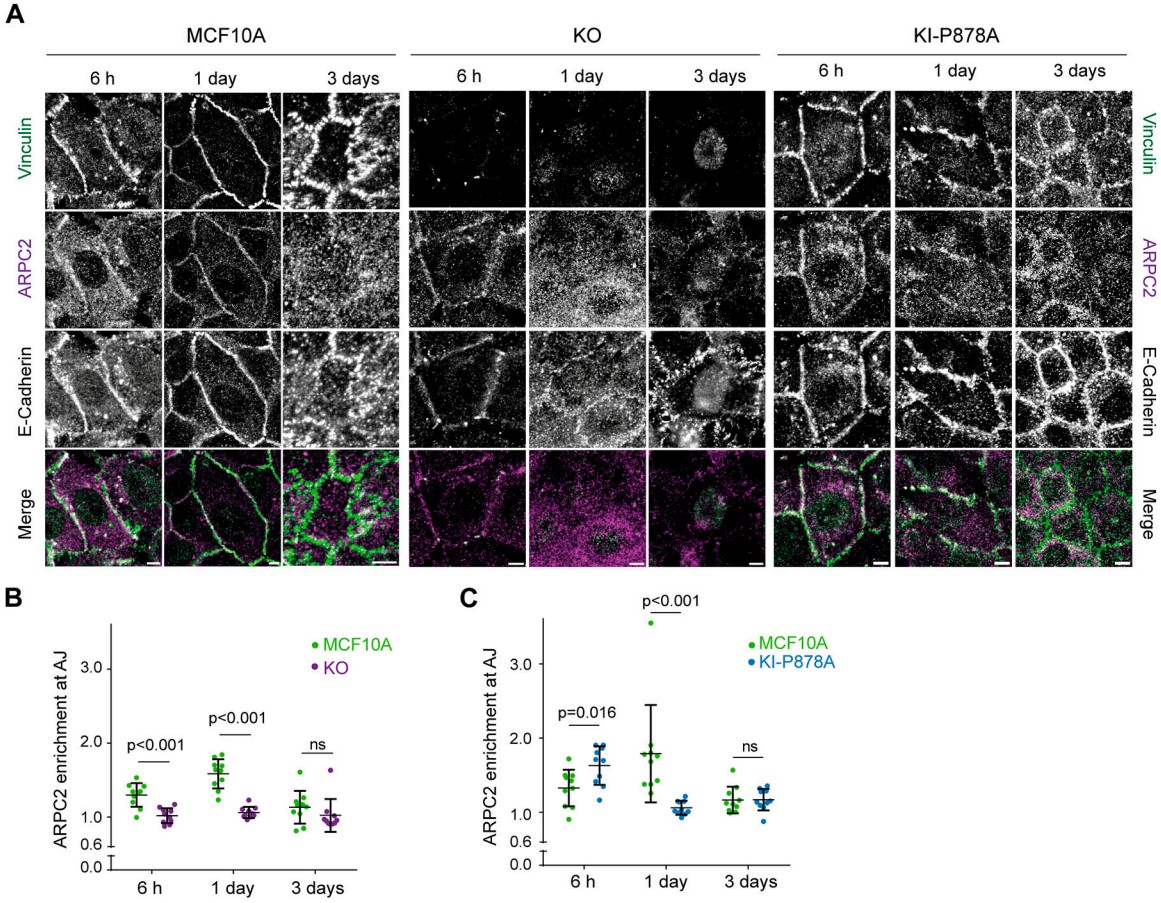

**Figure 7. Vinculin retains Arp2/3 at cell–cell junctions.**
**(A)** Staining of vinculin, ARPC2, and E-cadherin 6 h, 1 d, or 3 d after plating. Scale bar: 5 µm. Max z-projection from confocal microscopy. **(B, C)** Quantification of ARPC2 enrichment at adherens junctions in KO (B) and KI-P878A (C) cells. Mean ± SD, n = 10, *t* test. N = 3; 1 representative experiment shown.

Untransformed cells stop proliferating when confluent. This phenomenon commonly called "contact inhibition of proliferation" can be more precisely referred to as density dependence of cell cycle progression, because cells do not stop proliferating as soon as they touch each other, but rather enter less and less frequently into a new cell cycle, as the cell culture becomes denser. We observed that KO and KI-P878A cells reached a ~30% higher saturation density than parental MCF10A cells (Figs 8A and S9). We then measured the number of cycling cells by estimating the % of cells incorporating EdU, an analog of thymidine incorporated in DNA during the S phase. The % of cycling cells steadily decreased as a function of the cell density. KO cells behaved like parental cells, with a high cycling rate at low density and a low cycling rate at high density. However, at an intermediate density, KO cells were significantly more prone to enter into a new cell cycle than parental cells (Fig 8B). A similar behavior was observed for KI-P878A cells (Fig 8C), suggesting that the mere presence of vinculin was not sufficient to control contact inhibition; vinculin should also be able to interact with the Arp2/3. To confirm this point, we analyzed the density dependence of cell cycle progression on MCF10A cells expressing the vinculin linker. Cells expressing the dominant-negative construct exhibited significantly increased cycling even

at high cell density (Fig 8D). In contrast, the P878A mutation in the linker abolished this increased cycling, thus reinforcing the idea that vinculin controls cell cycle progression through its ability to interact with the Arp2/3 complex. When parental cells were treated with the Arp2/3 inhibitory compound CK-666, cell cycle progression was inhibited in a dose-dependent manner (Fig 8E). KO and KI-P878A cells, as well as MCF10A cells expressing the dominant-negative vinculin linker, also displayed a dose-dependent inhibition of cell cycle progression, but required more CK-666 to achieve the same level of inhibition (Fig 8E and F). Cell cycle progression thus appears to be inhibited by vinculin through its effect on Arp2/3 activity, in a manner similar to what occurs for membrane protrusions and persistence of single-cell migration.

## Discussion

The vinculin–Arp2/3 interaction was first assumed to be a transient regulated interaction that involved the canonical Arp2/3 complex (DeMali et al, 2002). However, alternative assemblies of so-called vinculin–Arp2/3 hybrid complexes have then been discovered

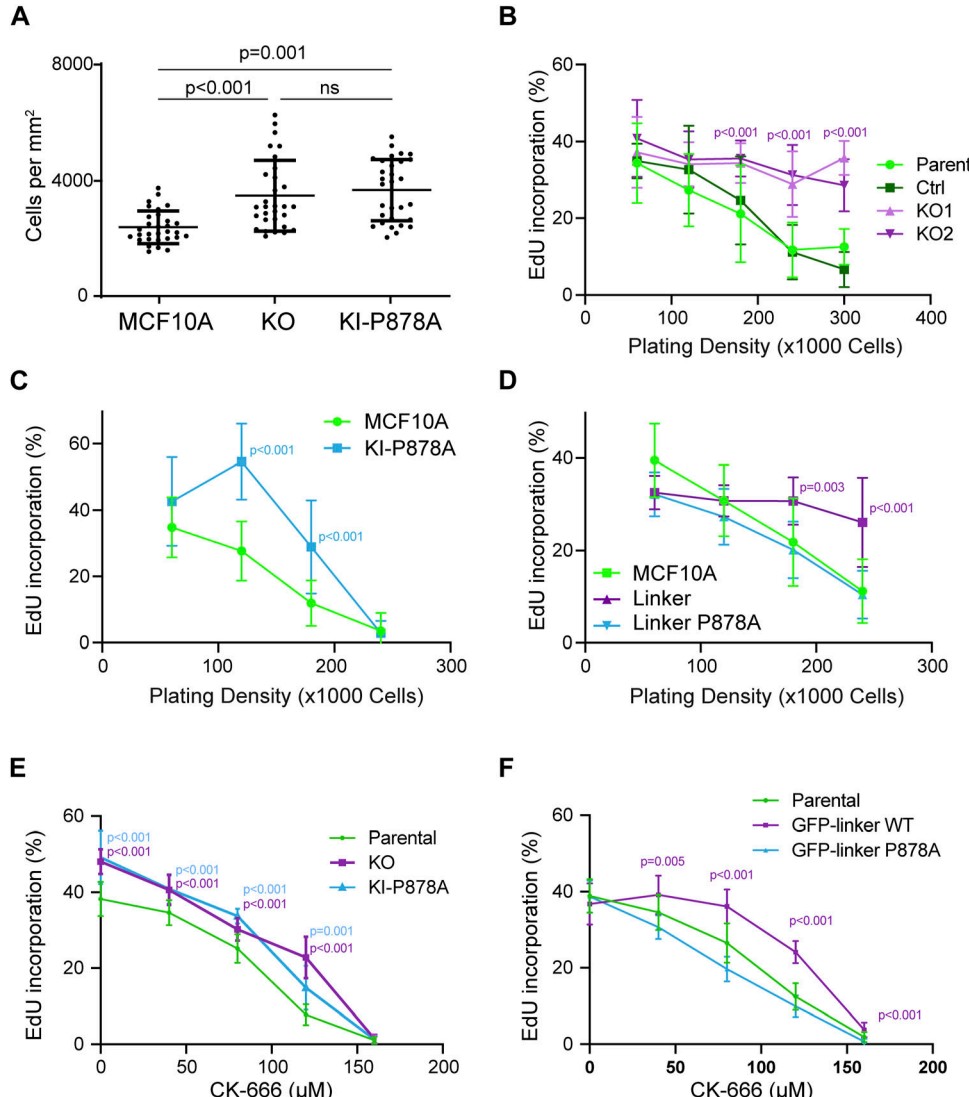

**Figure 8. Vinculin controls cell cycle progression through its interaction with the Arp2/3 complex.**
**(A)** Saturation density of KO and KI-P878A cells. Mean ± SD, n = 30, *t* test. N = 3 with similar results; pooled measurements from three independent repeats are plotted. **(B, C, D)** Cell cycle progression of KO- (B), KI-P878A– (C), and linker-expressing MCF10A cells (D). The percentage of cells incorporating EdU is represented as a function of cell density. N = 3 with similar results; pooled measurements from the three independent repeats are plotted. **(E, F)** Cell cycle progression of KO and KI-P878A cells (E), or linker-expressing MCF10A cells (F) plated at a density of $5 \times 10^4$ cells and treated with increasing doses of the Arp2/3 inhibitory compound CK-666. Mean ± SD, n = 8 fields of views per repeat with more than 15,000 cells in total, *t* test. N = 3 with similar results; pooled measurements from the three independent repeats are plotted. **(B, C, D, E, F)** *P*-values are shown only when both KOs and KI-P878A cells are different from both controls (B, C, E) and when the linker is significantly different from parental MCF10A (D, F).

(Chorev et al, 2014). Because Arp2/3 subunits analyzed in vinculin immunoprecipitates by DeMali belonged to both canonical and hybrid complexes, it was not known whether the two modes of binding existed or whether the vinculin–Arp2/3 interaction only involved assembly of hybrid complexes. In the present study, we unambiguously detected subunits that belong to the canonical Arp2/3 complex, but not to hybrid complexes, in vinculin-linker immunoprecipitates or at cell–cell junctions at a time when vinculin is solely responsible for Arp2/3 recruitment. Therefore, even if these experiments do not rule out the presence of vinculin–Arp2/3 hybrid complexes in MCF10A cells, they show that the interaction of the canonical Arp2/3 complex with vinculin through the vinculin linker does exist, as suggested by the original reference. We also favor the interpretation that the roles of the vinculin–Arp2/3 interaction we report here are due to an interaction of vinculin with the canonical Arp2/3 complexes, because these functions in membrane protrusion, migration persistence, and cell cycle progression were all previously ascribed to the

canonical Arp2/3 complex (Suraneni et al, 2012; Wu et al, 2012; Molinie et al, 2019).

Our strategy to compare the phenotypes of KO and KI-P878A cells allowed us to distinguish the mechanotransducer function of vinculin that only depends on the presence of vinculin from the vinculin functions that require both vinculin and its interaction with Arp2/3 (Fig 9). Among the Arp2/3-dependent functions of vinculin in MCF10A cells, we found an inhibition of branched actin assembly, membrane protrusions, and cell spreading. It was reported in the original article mapping the Arp2/3 binding site on vinculin that vinculin KO MEFs had impaired lamellipodia and cell spreading (DeMali et al, 2002). These two phenotypes were rescued by the expression of WT vinculin, but not by the P878A derivative. Our two studies thus implicate the same functions, but with opposite cellular effects. Similarly, the increased migration persistence we observed in vinculin KO MCF10A cells is in contrast to the decreased persistence observed upon vinculin depletion in MEF KO cells or siRNA-treated mammary carcinoma MDA-MB-231 cells, using

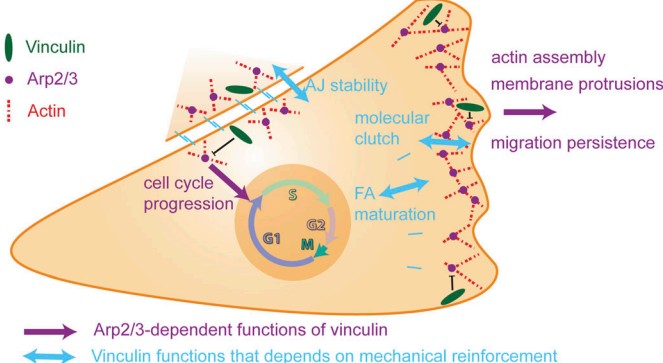

**Figure 9. Vinculin controls cell migration and cell cycle progression through its ability to interact with the Arp2/3 complex.**
The vinculin–Arp2/3 interaction antagonizes branched actin polymerization in membrane protrusion and inhibits migration persistence of single cells. The actin reinforcement provided by vinculin stabilizes cell–cell junctions. Subsequent vinculin-dependent recruitment of Arp2/3 at cell–cell junctions is likely to contribute to the Arp2/3-dependent role of vinculin in collective migration and density-dependent inhibition of cell cycle progression.

different assays (Thievessen et al, 2015; Rahman et al, 2016; Lee et al, 2019). Even if the migration phenotypes of vinculin-depleted cells strongly depend on cell types and precise assay conditions, in particular 2D versus 3D conditions (Fernández et al, 1993; Mierke et al, 2010; Thievessen et al, 2015), it is clear that untransformed epithelial cells, MCF10A cells, behave in a significantly different manner than the previously examined cell types. This kind of cell type–specific behavior has been recently described for vimentin expression, which has the opposite effects in MCF10A cells than in most other cell types and decreases migration persistence (Fokin et al, 2024).

Another surprising observation came from the examination of the molecular clutch that connects actin polymerization at the leading edge to cell adhesions to the substratum. In fibroblasts, vinculin has been implicated in mediating the molecular clutch that connects the retrograde flow of actin to cell adhesions to the substratum (Thievessen et al, 2013; Hirata et al, 2014). Because the lamellipodia mainly consist of branched actin, one can expect that vinculin–Arp2/3 interactions could be part of the molecular clutch. In epithelial MCF10A cells, we confirm the vinculin function in the molecular clutch, but exclude a role for vinculin–Arp2/3 interactions. In our vinculin KO and KI-P878A MCF10A cells, the release of branched actin inhibition can account for the increased persistence of single-cell migration. Moreover, the increased efficiency of membrane protrusions in vinculin KI-P878A cells compared with KO and parental MCF10A cells fully agrees with vinculin KI-P878A cells being slightly more persistent and more spread than vinculin KO cells.

Our epithelial cell system allowed us to examine the role of vinculin at cell–cell adhesions. We found that vinculin KO MCF10A cells had dramatically decreased stability of AJs when cells were embedded into soft 3D collagen gels, highlighting the interplay between adhesions to the substratum and to the neighboring cells. Vinculin belongs to both cell adhesions, the specific incorporation into cell–cell adhesions being determined by Abl-mediated phosphorylation of Y822 (Bays et al, 2014). Decreased cell–cell

adhesion upon vinculin KO was recently observed in the mouse 4T1 breast cancer line and murine skin cells (Biswas et al, 2021; DeWane et al, 2023) and is in line with the aberrant AJs between cardiac myocytes reported in heart-specific KO of vinculin in mice (Zemljic-Harpf et al, 2007). Vinculin is an essential component of cell–cell junctions that allows myosin-dependent tensile forces to develop mature junctions (Twiss et al, 2012). KI-P878A cells do not exhibit unstable AJs and even exhibit more stable AJs than parental cells. Disabling Arp2/3 binding thus appears to increase the function of vinculin in reinforcing AJs and weakening their plasticity. Increased pushing forces provided by branched actin at early cell–cell junctions can account for the increased junction stability of KI-P878A cells, if we assume that branched actin is later remodeled in linear arrays for myosin-mediated contractility. Indeed, branched actin is a poor substrate for myosin motors (Muresan et al, 2022), but GMF and coronin proteins were shown to debranch actin networks of lamellipodia (Cai et al, 2008; Haynes et al, 2015) and might play a similar role at cell–cell junctions.

In the 2D wound healing assay of MCF10A cells, vinculin KO cells were more efficient at closing the wound than parental cells, as previously reported using 4T1 cells (DeWane et al, 2023). KO MCF10A cells exhibited fast and directional migration toward the wound and transmitted the signal further back in the monolayer, indicating that the mechanotransduction of E-cadherin–dependent cell adhesions that vinculin provides (le Duc et al, 2010) is not essential to this transmission, and even rather inhibitory. These phenotypes are less prominent when vinculin is present, but impaired in its Arp2/3 interaction, indicating that the vinculin–Arp2/3 interaction is less critical for collective migration than for single-cell migration.

Cadherins at AJs were found to be associated with branched actin, which pushes membranes from neighboring cells against one another to initiate cell–cell adhesion or repair unzipped membranes because of excessive tension (Efimova & Svitkina, 2018; Li et al, 2020, 2021; Senju et al, 2023). We found that vinculin was essential to recruit the Arp2/3 complex at AJs, but not at an early time point, 6 h after cell plating, where not only Arp2/3 recruitment did not depend on vinculin, but also Arp2/3 recruitment increased in KI-P878A cells. These results show that vinculin plays an essential role in Arp2/3 recruitment at AJs, but that there are also other ways to recruit it. The α-catenin molecule, which recruits vinculin at AJs, also binds to the Arp2/3 and inhibits it, but this involves a free form of α-catenin that is not bound to E-cadherin and β-catenin (Drees et al, 2005; Benjamin et al, 2010). The nucleation-promoting factors, WAVE and N-WASP, recruit Arp2/3 and induce polymerization of branched actin at AJs (Kovacs et al, 2002; Verma et al, 2012; Rajput et al, 2013). Cortactin, which is now recognized as an Arp2/3 stabilizer of the branched junction of actin filaments (Gautreau et al, 2022), is also critical for Arp2/3 localization at AJs (Helwani et al, 2004; Han et al, 2014). These proteins are obvious candidates for the early vinculin-independent recruitment of Arp2/3 at AJs we observed here. However, their implication is difficult to test, because they also have a critical role in the formation and maintenance of AJs, the very structure where one would like to assess Arp2/3 recruitment.

Vinculin was found to control saturating cell density of cultures and regulate cell cycle progression as a function of cell density. This increased proliferation was observed in both KO and KI-P878A cells, indicating that this vinculin function strictly depends

on its ability to interact with the Arp2/3. We previously established that cell cycle progression in untransformed cells depends on cortical branched actin (Molinie et al, 2019). In fact, the Arp2/3-dependent functions of vinculin uncovered here, membrane protrusions, persistence of single-cell migration, and cell cycle progression, were previously shown to depend on the RAC1-WAVE-Arp2/3 pathway (Dang et al, 2013; Molinie et al, 2019; Simanov et al, 2021). Increased protrusions, increased persistence, and increased cycling observed in KO and KI-P878A MCF10A cells are phenotypes, which are all associated with increased branched actin, showing that the vinculin–Arp2/3 interaction should antagonize the formation of branched actin. This could originate from the destabilization of the branched actin network, similar to coronin, GMF, or cofilin (Cai et al, 2008; Chung et al, 2022). Alternatively, vinculin could sequester inactive Arp2/3 complexes or, like Arpin, prevent Arp2/3 activation (Dang et al, 2013; Fregoso et al, 2022). Nevertheless, we found that the vinculin linker that binds the Arp2/3 in the cell was not affecting Arp2/3 activity in vitro, in the pyrene–actin assay, suggesting that additional factors or post-translational modifications are probably involved in the cell. Future work should be aimed at deciphering the precise molecular mechanisms by which vinculin antagonizes the nucleation of branched actin networks by the Arp2/3 complex.

# Materials and Methods

## Cell culture and drugs

MCF10A cells were grown in DMEM/F12 medium supplemented with 5% horse serum, 20 ng/ml EGF, 10 µg/ml insulin, 100 ng/ml cholera toxin, 500 ng/ml hydrocortisone, and 100 U/ml penicillin. Medium and supplements were from Life Technologies and Sigma-Aldrich. Cells were incubated at 37°C in 5% $CO_2$. Cells were trypsinized (12605010; Gibco) and subcultured every 3 d. CK-666 (182515; Sigma-Aldrich) was used for Arp2/3 inhibition as stated.

## Plasmids, transfection, and isolation of stable cell lines

GFP-tagged proteins were expressed from a home-made vector, MXS AAVS1L SA2A Puro bGHpA EF1Flag GFP Blue SV40pA AAVS1R, that was previously described (Molinie et al, 2019). Vinculin linkers WT, P878A, ARPC1A, ARPC1B, ARPC5, and ARPC5 were inserted into this plasmid in place of the Blue cassette using Fse1 and Asc1 restriction sites. The P878A mutation was generated from the WT plasmid using QuikChange Lightning Mutagenesis Kit (Agilent) and primers (CCTAGGCCTCCACCAGCAGAGGAAAAGGATG, GTAGGAAAAGGAGACGACCACCTCCGGATCC).

Transfections of MCF10A cells were performed using Lipofectamine 3000 (Invitrogen). To obtain stable cell lines, MXS AAVS1 vectors were cotransfected with two TALEN constructs (#59025 and 59026; Addgene) inducing a double-strand break at the AAVS1 locus (González et al, 2014). Cells were selected with 1 µg/ml puromycin (ant-pr-1; InvivoGen) and pooled if the expression was homogeneous or cloned otherwise.

## Genome editing

To generate vinculin KO lines, MCF10A cells were transfected with a sgRNA (ATCGTGCGCGTATGAAACAC) targeting nucleotides 7–26 of the VCL coding sequence, corresponding to amino acids 3–9 of the vinculin protein, along with the purified Cas9 protein using the Lipofectamine CrisprMax kit (#CMAX00001; Thermo Fisher Scientific). Cells were then diluted and seeded in a 96-well plate at 1 cell/well. Wells containing two or more clones were not analyzed. 130 clones were screened by dot blot using anti-vinculin antibodies at 1:1,000 dilution. Cells with minimal signal on dot blots were further screened using Western blot, immunostaining, and sequencing to derive characterized KO clones.

To characterize the VCL mutations on the vinculin gene, base pairs 14–494 were amplified by PCR using DreamTaq (EP0702; Thermo Fisher Scientific) for 32 cycles with an annealing temperature of 58°C and the (TCTGTCTCTTCGCCGGTTC, AGCCTTTTTCAT-GACTGCTCC) primers. The PCR product was sequenced. When several sequences overlapped, PCR products were cloned into a blunt vector using the Zero Blunt PCR cloning kit (#K270040; Thermo Fisher Scientific) to sequence the two alleles independently.

To obtain the vinculin KI-P878A line, MCF10A cells were cotransfected by electroporation with a Cas9-expressing plasmid (CMV hSpCas9 bGH pA), a plasmid expressing the ATP1A1 sgRNA (Agudelo et al, 2017), a pRG2(-GG) plasmid expressing the VCL sgRNA (GCCTCCACCACCAGAGGAAA), and a single-stranded 87-bp repair oligonucleotide (base pairs 110878–110964 in the VCL gene). Colonies resistant to ouabain (0.5 µM, 03125; Sigma-Aldrich) were cloned using dilution and screened by PCR using DreamTaq (EP0702; Thermo Fisher Scientific) for 32 cycles with an annealing temperature of 52°C and the (GGTGACGATC-GAAAAAC, TATTGGCAACACAGGAACC) primers, followed by PvuII restriction.

## Antibodies

Antibodies used for immunostaining and Western blots were anti-vinculin (#V9131; Sigma-Aldrich), anti-α-catenin (#C2081; Sigma-Aldrich), anti-E-cadherin (#MABT26; Merck), anti-paxillin (#GTX125891; GeneTex), anti-ARPC2 (#07-227-I; Millipore), anti-ARPC1B (#HPA004832; Sigma-Aldrich), and anti-ARPC3 (#HPA006550; Sigma-Aldrich). For immunostaining, secondary antibodies for anti-mouse-647 (#A21236; Life Technologies) and anti-rabbit-405 (#A34556; Life Technologies) were used along with Acti-stain 555 (Cytoskeleton).

## Immunoprecipitation and Western blot

Two 15-cm dishes of MCF10A cells were lysed 1 d after seeding and scraped off in 50 mM Hepes, pH 7.7, 10 mM EDTA, 50 mM KCl, 1 mM $MgCl_2$, and 1% NP-40 supplemented with a protease inhibitor cocktail (1:10,000; Roche). Cell lysates were centrifuged at 10,000g for 10 min. The supernatant was incubated at 4°C for 1 h with GFP-Trap Agarose beads (ChromoTek) and washed four times. Lysates and immunoprecipitates were analyzed by Western blot.

## Cell attachment assay

Cell attachment assays were performed as previously described (Humphries, 2009). Briefly, 25 × 10³ cells were seeded in 96-well plates coated with either fibronectin, and saturated with heat-denatured BSA, or heat-denatured BSA alone. After incubation at 37°C in a $CO_2$ incubator for 15 min, cells were washed, fixed, and stained with a 0.1% (wt/vol) crystal violet. The amount of cells stained by crystal violet in each condition was determined by reading OD at 570 nm.

## Immunofluorescence and image analysis

To image cells, 15,000 cells (to obtain individual cells) or 1.5 × 10⁶ cells (to obtain a monolayer) were plated on 22 × 22 mm coverslips that were coated with fibronectin (10 $\mu$g/ml, F1141; Sigma-Aldrich) and fixed after 1 d (unless otherwise stated) with 4% PFA for 15 min. Cells were permeabilized in 0.2% Triton X-100 and blocked in 10% FBS in PBS. Cells were stained in a 1:200 dilution of first primary and then secondary antibodies along with Acti-stain 555. Coverslips were mounted in Dako mounting medium. For immunofluorescence of cells embedded in a 3D collagen matrix, 15,000 cells were plated into microchambers (#80826; ibidi) on a 3.5 mg/ml collagen type 1 (#354236; Corning) matrix in DMEM:F12 supplemented with FBS (10%). After cells were attached, another layer of collagen was added on top and cells were incubated in culture medium for 1 d before imaging. Samples were processed as described in Artym and Matsumoto (2010). Samples were imaged using an SP8 laser-scanning confocal microscope (Leica). Images were analyzed using Fiji. To measure cell spreading, the cell area was quantified on images by thresholding (Triangle algorithm) in Fiji.

To quantify vinculin recruitment at AJs and FAs, the thresholded $\alpha$-catenin or paxillin staining, respectively, was used to generate a mask in which the fluorescence intensity of vinculin was quantified. Lengths of focal adhesions were manually measured using paxillin staining. To quantify Arp2/3 enrichment at the edge of lamellipodia, LineScan analyses were performed on single-plane confocal microscopy images of fixed, stained cells, which were obtained and analyzed as previously described (Dang et al, 2013; Molinie et al, 2019). To quantify Arp2/3 and vinculin enrichment at AJs, a line was manually drawn along the junction labeled with anti-E-cadherin. The width of the line was then increased iteratively to measure total fluorescence intensity at increasing distances from the junction. Intensity at a distance n from the junction corresponded to (total intensity of line width$_n$)—(total intensity of line width$_{n-1}$). Values were finally normalized to the average fluorescence intensity at a line ≈ 12 $\mu$m from the junction, corresponding to intensity in the cytoplasm. Enrichment of E-cadherin at AJs was measured similarly using phalloidin staining as a reference for cell–cell contacts.

## Live-cell imaging

For 2D cell migration assays, 15,000 cells were seeded 1 d before imaging on microslides (#80826; Ibidi) coated with fibronectin. For 3D migration of cells embedded in a collagen matrix, 15,000 cells were plated into microchambers (#80826; Ibidi) on a 3.5 mg/ml collagen type 1 (#354236; Corning) matrix in DMEM:F12 supplemented with FBS (10%). After cells were attached, another layer of collagen was added on top and cells were incubated in culture medium for 1 d before imaging. Cells were imaged on an Axio Observer Z1 microscope (Zeiss) equipped with an Orca-R2 CCD camera (Hamamatsu) and controlled by AxioVision software (Zeiss). Images were acquired at 5-min (2D migration) or 10-min intervals (migration in a 3D collagen matrix) for 24 h. Cell–cell junction disassembly events were counted manually at each time point. Analysis of migration persistence was performed as previously described (Gorelik & Gautreau, 2014).

For TIRF-SIM imaging, 100,000 mCherry–actin-transfected cells were plated 1 d before imaging on glass-bottom dishes (P35G-0.170-14-C; MatTek Corp) coated with fibronectin. To image actin flows in lamellipodia, images were acquired at 2-s intervals for 2 min using 3 phase-shifted angles, each with three fringe patterns, on the DeltaVision OMX SR (GE Healthcare) microscope. High-resolution images were reconstructed, and two-color images were aligned using softWoRx (Applied Precision). Kymographs were generated in Fiji using manually drawn lines that followed the direction of actin retrograde flow and the Multi Kymograph tool. Protrusion speed and rearward flow are given by the tan of the angle made with the time axis in kymographs. Protrusion speed and rearward flow values have opposite signs, but absolute values are given for clarity. The actin assembly rate is the sum of protrusion speed and rearward flow.

## Analysis of wound healing experiments

80,000 cells were plated on microslides (#80826; Ibidi) coated with fibronectin within inserts (#80209; Ibidi) 1 d before the experiment. Inserts were removed, and time-lapse acquisitions were performed at 10-min intervals during 24 h on an Axio Observer Z1 microscope (Zeiss) equipped with a 10x objective and an Orca-Flash4.0 v3 camera (Hamamatsu) and controlled by MicroManager 2.0 software. Fields focused on only one edge of the wound were acquired, and average leading-edge positions perpendicular to the axis of the wound gave leading-edge progression. The velocity field was obtained by PIV (Petitjean et al, 2010) using the PIVlab software package (Thielicke & Sonntag, 2021) for MATLAB (The MathWorks). The window size was set to 32 pixels, that is, 23.75 $\mu$m with a 0.75 overlap between windows. A time sliding window averaging velocity fields over 40 min (equal to four frames) was used. Spurious vectors were filtered out by their amplitude and were replaced by inter-polated velocities from neighboring vectors. The local order parameter was calculated using a previously published code (Deforet et al, 2012).

## Cell cycle and proliferation

To quantify saturation density, 2 × 10⁶ cells were plated on fibronectin-coated coverslips in six-well plates. Cells were fixed 4 d after seeding, and nuclei were stained with DAPI. To perform the EdU incorporation assay, cells were seeded on fibronectin-coated coverslips (12 mm) for 1 d. Cells were incubated with 10 $\mu$M EdU for 1 h before fixation in 4% PFA for 15 min and permeabilized in 1% Triton X-100 for 5 min. EdU was labeled with the Alexa Fluor 488 Click-iT EdU Imaging Kit (#C10337; Thermo Fisher Scientific)

according to the manufacturer's instructions, and nuclei were labeled with DAPI (Thermo Fisher Scientific). Images were acquired on an inverted microscope (Olympus IX83) using a 20x objective (NA 0.5) equipped with an Orca-Flash4.0 V3 camera (Hamamatsu) and controlled by MicroManager 2.0, and analyzed using a custom script in Fiji to count DAPI- or EdU-positive cells. The percentage of cells in the S phase was scored as the ratio of EdU-positive nuclei to DAPI-stained nuclei in segmented images. For each condition, at least 5,000 cells were counted.

### Statistical analysis

For all $t$ tests, populations were first tested for the Gaussian distribution using a Shapiro–Wilk test with a $\alpha$-value of 0.05. If both populations were Gaussian-distributed, the difference between means was tested using Welch's $t$ test, and if one or both of the populations were non–Gaussian-distributed, the difference between means was tested using a Mann–Whitney $U$ test. The focal adhesion size, protrusion speed, and frequency of cell junction disassembly were tested with a one-tailed distribution as the lower limit of measurement was close to a lower bound of 0. All other $t$ tests assumed a two-tailed distribution. Analysis of migration persistence was performed as previously described (Gorelik & Gautreau, 2014). Exponential decay and plateau fit ($y = (1 - b) * e^{-\frac{t}{a}} + b$) were performed for all individual cells. Coefficients were then compared using one-way ANOVA. Statistical analysis was performed in R using linear mixed-effect models to take into account the resampling of the same statistical unit as previously described (Polesskaya et al, 2022 Preprint).

# Supplementary Information

# Acknowledgements

This work was supported by grants from Agence Nationale de la Recherche (ANR-20-CE13-0016, ANR-22-CE44-0006, and ANR-22-CE13-0041), Fondation ARC pour la Recherche sur le Cancer (ARC PJA 2021 060003815), Institut National du Cancer (INCA_16712), Federal Territory "Sirius," N 22-03, dated 27.09.2024, and Engineering and Physical Sciences Research Council (EPSRC, EP/V043498/1).

## Author Contributions

J James: formal analysis, investigation, and writing—original draft.
AI Fokin: investigation.
DY Guschin: investigation.
H Wang: investigation.
A Polesskaya: investigation.
SN Rubtsova: investigation.
C Le Clainche: conceptualization and supervision.
P Silberzan: conceptualization and supervision.
AM Gautreau: conceptualization, supervision, validation, and writing—original draft, review, and editing.
S Romero: conceptualization, formal analysis, supervision, validation, investigation, and writing—original draft, review, and editing.

## Conflict of Interest Statement

The authors declare that they have no conflict of interest.

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
