## [Reviewer comments · Life Science Alliance]

Life Science Alliance

Vinculin-Arp2/3 Interaction Inhibits Branched Actin Assembly to Control Migration and Proliferation

John James, Artem Fokin, Dmitry Guschin, Hong Wang, Anna Poleskaya, Svetlana Rubtsova, Christophe Le Clainche, Pascal Silberzan, Alexis Gautreau, and Stéphane Romero

DOI: <https://doi.org/10.26508/lsa.202402583>

Corresponding author(s): Stéphane Romero, École Polytechnique

Review Timeline:

Submission Date:	2024-01-09
Editorial Decision:	2024-02-16
Revision Received:	2024-10-07
Editorial Decision:	2024-10-25
Revision Received:	2024-11-04
Accepted:	2024-11-05

Transaction Report:

February 16, 2024

Re: Life Science Alliance manuscript #LSA-2024-02583-T

Dr. Stéphane Romero
Collège de France
CIRB, Inserm U1050, CNRS UMR7241
11, Place Marcelin Berthelot
Paris 75005
France

Dear Dr. Romero,

Thank you for submitting your manuscript entitled "Vinculin-Arp2/3 Interaction Inhibits Branched Actin Assembly to Control Migration and Proliferation" to Life Science Alliance. The manuscript was assessed by expert reviewers, whose comments are appended to this letter. We invite you to submit a revised manuscript addressing the Reviewer comments.

Thank you for this interesting contribution to Life Science Alliance. We are looking forward to receiving your revised manuscript.

Sincerely,

B. MANUSCRIPT ORGANIZATION AND FORMATTING:

Reviewer #1 (Comments to the Authors (Required)):

This study investigates the cellular functions of the vinculin-arp2/3 interaction. The authors utilize MCF10 cells in which they either i) knockout Vinculin, ii) introduce a P878A mutation in the linker region of (endogenous) vinculin that impairs Arp2/3 binding, iii) stably overexpress the linker region, with or without the P878A mutation. Comparison of these different cells indicates that the interaction of vinculin with arp2/3 antagonizes branched actin polymerization at membrane protrusions and decreases migration persistence of single cells. In contrast, regulation of focal adhesion maturation and formation of stable cell-cell junctions requires Vinculin but not its interaction with Arp2/3. The authors further monitor the recruitment of Arp2/3 to cell-cell junctions during their formation and show junctional Arp2/3 recruitment 1 day after plating cells relies on its binding to Vinculin (and initial recruitment of Arp2/3 does not). Finally, the authors show that impairing Arp2/3 binding to vinculin affects collective migration and cell cycle progression. Altogether, this work provides new insights into the contribution of its interaction with Arp2/3 in vinculin-dependent single-cell and multicellular processes.

Unfortunately, the manuscript lacks a clear connection between its various findings, dampening my enthusiasm for this study. Moreover, I have significant concerns about the incompleteness of the presented analyses and the absence of sufficient controls to substantiate the drawn conclusions.

Main concerns

1. My primary concern centers around the lack of coherence among the various observations in this study, limiting the advancement of our understanding of the Vinculin-Arp2/3 interaction. For instance, Figures 1-3 demonstrate the impact of introducing the P878A mutation on branched actin assembly in protrusion, cell spreading, and migration persistence, but do not investigate the distribution of the Arp2/3 complex under these conditions. On the other hand, the authors explore how the junctional localization of Arp2/3 involves its interaction with Vinculin (Fig. 6), yet in this case the functional consequences of this localization (e.g. actin organization) remain unexplored. The study further describes the influence of the P878A mutant on collective migration and cell cycle progression, without establishing any connection to Arp2/3 localization or actin organization. Thus, while the manuscript provides an overview of cellular phenotypes affected by the P878A mutant (or overexpression of the linker, and compared to vinculin knockout), it lacks fundamental insights into how the Vinculin-Arp2/3 interaction might play a role in these processes (through localizing arp2/3? Inhibiting arp2/3?). Significantly, despite the absence of clear connections between experiments, the authors draw robust conclusions, as evident in the concluding model. For instance, the authors assert that "vinculin-dependent recruitment of Arp2/3 at cell-cell junctions contributes to collective migration and density-dependent inhibition of cell cycle progression." However, the data supporting the claim that junctional recruitment of Arp2/3 contributes to these processes are absent (of note, it is also unclear how this conclusion would fit with the transient role of Vinculin in recruiting Arp2/3 to cell-cell contacts during junction formation and not in mature junctions, fig 6). Altogether, this renders the conclusions on the function of Arp2/3 in the described processes speculative at best.

2. Throughout the entire manuscript, the authors only present the analysis of a single representative experiment. To enable the interpretability and reliability of the data and conclusions, it is imperative to include data and statistical analyses from all independent experiments rather than selectively showcasing one.

3. The conclusion that the linker domain behaves dominant negatively is based on phenotypes of cells with the linker being most comparable knockout cells. Sufficiently convincing evidence to support this conclusion is lacking; the authors should at least test the presumable absence of effects of linker overexpression in Vinculin knockout cells. Ideally, the authors should also test whether the linker disrupts the endogenous Vinculin-Arp2/3 binding. Importantly, the authors should show whether levels of endogenous Vinculin are comparable between control cells and cells stably expressing the linker and linker P787A.

4. Further characterization of the KO and KI clones is essential, particularly given the discordant findings compared to earlier work (as described in the discussion). The authors introduce an indel in exon 1 to generate knockout cells: can the authors exclude a shorter, truncated product is expressed (note that this cannot be concluded with the Western Blot using the hVin-1 antibody for which the epitope is in the head domain of vinculin)? Regarding KI cells, the authors should show (protein) expression levels of this mutant protein expression and whether this is comparable to wildtype vinculin.

5. The effect of Vinculin KO on cell-cell junction formation is only observed in 3D collagen gels and not on 2D substrates. The authors subsequently conclude that the effect on junction stability is not due to regulation of E-cadherin-dependent adhesion, because E-cadherin was properly recruited to cell-cell junctions in KO cells (Suppl Fig 3). However, this was tested on 2D substrates and should be tested under the same conditions as when junctional defects are seen. Moreover, Fig. 6 shows a clear reduction of E-cadherin levels at junction upon Vinculin depletion, can the authors explain this discrepancy?
6. The authors conclude that the vinculin-Arp2/3 interaction regulates cell-cell junction plasticity through Arp2/3 recruitment (e.g. p4). It is unclear on which data this is based, because the lack of junctional Arp2/3 recruitment in P878A mutant cells (fig. 6) does not lead to any observed changes in junctional phenotype.
7. In Fig 1a, the authors should show the levels of ARPC1B and ARPC3 in the total lysates in all conditions.
8. The authors conclude that their data demonstrate vinculin's interaction with the canonical Arp2/3 complex in MCF10A cells. I recommend nuancing this conclusion, as the interaction is only demonstrated with the truncated linker and not full-length vinculin. Furthermore, enrichment of these canonical Arp2/3 proteins at cell-cell junctions (Suppl. Fig 5) does not demonstrate an interaction with vinculin.

Reviewer #2 (Comments to the Authors (Required)):

This is an interesting study focusing on the cellular roles of vinculin - Arp2/3 complex interactions. By careful analysis of wild-type, vinculin knockout, and vinculin knock-in cells (with a mutation in the Arp2/3-binding site), as well as by over-expressing wild-type and mutant versions of the vinculin 'linker region', the authors provide evidence that vinculin-Arp2/3 interactions inhibit actin polymerization and membrane protrusions, whereas the molecular clutch coupling branched actin to cell adhesions is not dependent on vinculin's interaction with the Arp2/3 complex. Moreover, they provide evidence that vinculin recruits Arp2/3 to cell-cell junctions and inhibits cell cycle in an Arp2/3-dependent manner.

The authors have used several cutting-edge approaches in these studies, and the data appear convincing and provide important new information on the interplay between vinculin and the Arp2/3 complex. However, there are few relatively minor points that should be addressed to further strengthen this manuscript.

1. The introduction to the vinculin - Arp2/3 interaction (on page 4 of the manuscript) was bit superficial, and the authors should more precisely explain here what is the evidence (based on the previous publications) for the direct interaction between vinculin and Arp2/3 complex, and what is already known/reported about the biochemical effects of vinculin on the Arp2/3 complex and Arp2/3-nucleated actin filament networks.
2. Similarly, the authors should at least speculate in the 'Discussion' what are the possible molecular mechanisms by which vinculin affects the localization and /or activity of the Arp2/3 complex. Because based on Fig. S2 vinculin does not affect VCA-induced actin filament assembly by the Arp2/3 complex, is it possible that vinculin would affect the stability of Arp2/3-nucleated branches through GMF or cortactin? Alternatively, vinculin could sequester Arp2/3 complex from the membrane-associated NPFs to inhibit Arp2/3-catalyzed actin filament assembly in cells. To examine the latter option, the authors could just simply analyze the localization of the Arp2/3 complex in their wild-type, vinculin knockout, and vinculin knock-in cells (to see if e.g. in the knock-in cells, the intensity of Arp2/3 would be decreased at the edges of lamellipodia).
3. The image quality was not particularly good (especially in Figs. 1, and 2), and thus the authors may consider providing better quality immunofluorescence images and time-lapse images from the movies.
4. The analysis of protrusion speed/rearward flow/actin assembly rate/protrusion efficiency was bit confusing. It appears that the authors have just analyzed protrusion speed and actin rearward flow from the obtained kymographs, and thus plotting also the actin assembly rate and protrusion efficiency in the figures is not particularly informative.
5. What are the 'control cells' in Fig. 2D & E, Fig 3E, and Fig. 4 B & G? How do these differ from the wild-type MCF10A cells? This should be specified.
6. The sentence: 'We found that antibiotic-selected MCF10A cells down-regulated the exogenous expression of tagged full-length vinculin in an increasing number of cells over time, but not when the construct was limited to the vinculin linker that connects the head to the tail and which contains the Arp2/3 binding site.' in the beginning of 'Results' is difficult to follow, and some additional information or explanation is needed here for clarification.
7. Scale bar is missing from Fig. S6.

Reviewer #3 (Comments to the Authors (Required)):

In this manuscript, James et al. investigate the interaction between vinculin and the Arp2/3 complex, and how this interaction regulates cell migration and proliferation. Contrary to previous studies, the authors find that the vinculin-Arp2/3 interaction inhibits the activity of Arp2/3. While the mechanism of inhibition by the vinculin linker remains largely unexplored, the presented data supports this notion. The authors might consider speculating on the potential mechanisms through which the vinculin linker inhibits Arp2/3 activity to further strengthen their manuscript. The brief mention of post-translational modification invites further speculation which could enrich the discussion.

Additionally, the use of epithelial cells instead of fibroblasts to demonstrate the cellular-context-dependent role of vinculin is a key and novel finding of this study. The authors could elaborate on the specific differences between epithelial cells and fibroblasts that account for their observed behaviors.

Minor concerns below need to be addressed or clarified for the final version of the manuscript:

1. Data for knock-out (KO) and knock-in (KI) are often presented separately, making it challenging to compare and analyze the data holistically. It may be beneficial to combine the data into a single graph. Additionally, the term "KI" should be clarified, especially in the figures; for instance, using "KI-P878A" instead of "KI" to denote vinculin P878A knock-in.
2. On page 7, the statement that "Arp2/3 binding is not required for the focal adhesion (FA)-related functions of vinculin" is ambiguous. The localization and length of FAs may not sufficiently conclude their adhesive function. Including assays for adhesion strength to the extracellular matrix may help clarify the functional roles of this interaction.
3. On page 8, the authors state that cell-cell interactions were "not obviously affected in KO cells" on 2D surfaces. However, Supplementary Movie 7 shows more single cells at the leading wound edge in KO cells, which aligns with their 3D data. Discussing this observation could support their model.
4. The phrase "the mechanotransducer function of vinculin" on page 9, and the deduction of vinculin's mechanosensitive function from their data, are unclear. It may be more accurate to suggest that the vinculin-Arp2/3-independent function is necessary for the stability of cell-cell junctions.
5. The sentence on page 9, "In contrast, the interaction of vinculin with Arp2/3 is not required for the stability of cell-cell junctions, but rather enhances cell-cell adhesions," is confusing. Given the complexity of 3D cell migration, vinculin might play multiple roles, such as in 3D cell-ECM and cell-cell interactions, which could differ from their 2D counterparts. The evidence based on dissociation events might not sufficiently demonstrate the strength of cell-cell adhesion. The authors could clarify or reword this sentence to avoid confusion.

We sincerely thank the reviewers for their careful assessment of our work. Their questions and suggestions have helped us to improve the quality of our manuscript. Our answers are displayed in blue and cited references can be found at the end of this letter.

Reviewer #1 (Comments to the Authors (Required)):

This study investigates the cellular functions of the vinculin-arp2/3 interaction. The authors utilize MCF10 cells in which they either i) knockout Vinculin, ii) introduce a P878A mutation in the linker region of (endogenous) vinculin that impairs Arp2/3 binding, iii) stably overexpress the linker region, with or without the P878A mutation. Comparison of these different cells indicates that the interaction of vinculin with arp2/3 antagonizes branched actin polymerization at membrane protrusions and decreases migration persistence of single cells. In contrast, regulation of focal adhesion maturation and formation of stable cell-cell junctions requires Vinculin but not its interaction with Arp2/3. The authors further monitor the recruitment of Arp2/3 to cell-cell junctions during their formation and show junctional Arp2/3 recruitment 1 day after plating cells relies on its binding to Vinculin (and initial recruitment of Arp2/3 does not). Finally, the authors show that impairing Arp2/3 binding to vinculin affects collective migration and cell cycle progression. Altogether, this work provides new insights into the contribution of its interaction with Arp2/3 in vinculin-dependent single-cell and multicellular processes.

Unfortunately, the manuscript lacks a clear connection between its various findings, dampening my enthusiasm for this study. Moreover, I have significant concerns about the incompleteness of the presented analyses and the absence of sufficient controls to substantiate the drawn conclusions.

Main concerns

1. My primary concern centers around the lack of coherence among the various observations in this study, limiting the advancement of our understanding of the Vinculin-Arp2/3 interaction. For instance, Figures 1-3 demonstrate the impact of introducing the P878A mutation on branched actin assembly in protrusion, cell spreading, and migration persistence, but do not investigate the distribution of the Arp2/3 complex under these conditions.

On the other hand, the authors explore how the junctional localization of Arp2/3 involves its interaction with Vinculin (Fig. 6), yet in this case the functional consequences of this localization (e.g. actin organization) remain unexplored.

The study further describes the influence of the P878A mutant on collective migration and cell cycle progression, without establishing any connection to Arp2/3 localization or actin organization. Thus, while the manuscript provides an overview of cellular phenotypes affected by the P878A mutant (or overexpression of the linker, and compared to vinculin knockout), it lacks fundamental insights into how the Vinculin-Arp2/3 interaction might play a role in these processes (through localizing arp2/3? Inhibiting arp2/3?).

Significantly, despite the absence of clear connections between experiments, the authors draw robust conclusions, as evident in the concluding model. For instance, the authors assert that "vinculin-dependent recruitment of Arp2/3 at cell-cell junctions contributes to collective migration and density-dependent inhibition of cell cycle progression." However, the data supporting the claim that junctional recruitment of Arp2/3 contributes to these processes are absent (of note, it is also unclear how this conclusion would fit with the transient role of Vinculin in recruiting Arp2/3 to cell-cell contacts during junction formation and not in mature junctions, fig 6). Altogether, this renders the conclusions on the function of Arp2/3 in the described processes speculative at best.

Some of these criticisms are constructive, but others are unjustified.

The logical flow of our manuscript does not have any flaw. All 2D multicellular assays were performed in strictly identical experimental conditions (cell density, plating time), therefore conclusions drawn from one assay applied for another one. We report vinculin KO phenotypes in MCF10A cells, and only a subset of these phenotypes is found in the biallelic knock-in of a point mutation that impairs Arp2/3 interaction. The latter are simply interpreted as the functions that require the vinculin-Arp2/3 interactions, among the many functions of vinculin. We can even add that this level of demonstration is very rarely achieved in Life Science Alliance publications. If CRISPR-mediated KO have become common, knock-in are much more difficult and are usually obtained on a single allele to tag the endogenous protein with GFP for example. Here we managed to introduce a single point mutation in the two alleles. We believe that this achievement should be acknowledged and participate to the level of demonstration we achieve here. Previous literature on vinculin employed plasmid-based overexpression to express point mutations.

This being said, we agree with the reviewer that Arp2/3 distribution should have been examined in the initial submission. We have now obtained the distribution of Arp2/3 at the edge of lamellipodia of migrating cells for WT, KO and KI cell lines, and have included these new data in **Fig.3g-j**.

In the last point, the reviewer questions the link between the lack of arp2/3 recruitment at junctions and the defects in collective migration. The reviewer is right in terms of formal logics. However, we should be able to express this logical hypothesis like anyone else on less elaborate mutants. If we observe a proximal defect (lack of Arp2/3 recruitment, due to the mutation), it is highly likely to be linked to the distal effect (effect on collective migration) as a functional consequence. This is the most parsimonious hypothesis, to link these two observations, and it should be exposed to the reader for clarity. We have considerably toned down the way we express it, but this idea is still expressed in the revised version of our manuscript.

2. Throughout the entire manuscript, the authors only present the analysis of a single representative experiment. To enable the interpretability and reliability of the data and conclusions, it is imperative to include data and statistical analyses from all independent experiments rather than selectively showcasing one.

We now provide throughout all the manuscript data of the pooled independent repeats and provide as supplementary data the representative individual experiments (see **FigS1** and **FigS4**). Our conclusions remain unchanged.

The only exception is **Fig.7b,c** where independent experiments cannot be pooled, because ARPC2 immunofluorescence gave a variable overall staining, but each independent experiment support the conclusion that vinculin is required to recruit Arp2/3 at AJs, and the vinculin-Arp2/3 interaction is required to maintain Arp2/3 there. In the pooled data (**FigS7a**), the tendency of each individual repeats is conserved, although the differences of pooled ARPC2 levels are lower between WT and KO cells, for the reason explained above.

3. The conclusion that the linker domain behaves dominant negatively is based on phenotypes of cells with the linker being most comparable knockout cells. Sufficiently convincing evidence to support this conclusion is lacking; the authors should at least test the presumable absence of effects of linker overexpression in Vinculin knockout cells. Ideally, the authors should also test whether the linker disrupts the endogenous Vinculin-Arp2/3 binding. Importantly, the authors should show whether levels of endogenous Vinculin are comparable between control cells and cells stably expressing the linker and linker P787A.

We respectfully disagree with some of these assumptions. The phenotype of vinculin KO is increased protrusion and cell migration. The expression of the linker in parental cells induces the same phenotype. The reviewer suggests to over-express the linker in Vinculin KO cells but obviously, that could only lead to the same phenotype. We can envision an additive effect but mechanistically we do not see how it could rescue the phenotype.

Yet we agree with the reviewer that expression of the linker can be dominant negative by titrating an important partner, here the Arp2/3, from the endogenous protein. We have attempted to immunoprecipitate endogenous vinculin, but this immunoprecipitation was not efficient enough (no enrichment in IP compared with lysates) to detect associated Arp2/3 (**Fig.R1**). This experiment nonetheless shows in total lysates that there are similar levels of vinculin and Arp2/3. This is also shown in the revised **Fig.1**.

Figure removed per authors' request by LSA Editorial Staff.

4. Further characterization of the KO and KI clones is essential, **particularly given the discordant findings compared to earlier work** (as described in the discussion). The authors introduce an indel in exon 1 to generate knockout cells: can the authors exclude a shorter, truncated product is expressed (note that this cannot be concluded with the Western Blot using the hVin-1 antibody for which the epitope is in the head domain of vinculin)? Regarding KI cells, the authors should show (protein) expression levels of this mutant protein expression and whether this is comparable to wildtype vinculin.

For the last point first, we apologize to have overlooked this essential piece of information. We have now provided the western blot showing the similar expression levels of WT vinculin and P878A vinculin in parental and KI cells (**Fig.2h**). The level of expression of the two proteins is similar.

In the genetic characterization of the KO clones, the frameshift is generated after the 3rd codon, so there is only the first 3 amino-acids in common with vinculin in these KO clones. This information was already present in the original manuscript, but probably missed by the reviewer. Thus, this excludes the reviewer's hypothesis about a shorter truncated product inducing the different phenotypes observed in vinculin KO cells.

We understand that the reviewer is not at ease with the fact that our MCF10A model system revealed vinculin functions opposite to what was previously described in fibroblasts. This is not the first time that our model system displays behavior opposite to the expected. For example, we recently published that vimentin expression in MCF10A is associated with decreased migration persistence instead of increased persistence (Fokin et al., 2024). We believe that this variability is quite interesting in itself and that the field should know about it, especially since our system allowed us here to go significantly further in identifying the specific functions of vinculin that require its interaction with Arp2/3.

5. The effect of Vinculin KO on cell-cell junction formation is only observed in 3D collagen gels and not on 2D substrates. The authors subsequently conclude that the effect on junction stability is not due to regulation of E-cadherin-dependent adhesion, because E-cadherin was properly recruited to cell-cell junctions in KO cells (Suppl Fig 3). However, this was tested on 2D substrates and should be tested under the same conditions as when junctional defects are seen. Moreover, Fig. 6 shows a clear reduction of E-cadherin levels at junction upon Vinculin depletion, can the authors explain this discrepancy?

We agree with the reviewer that the E-cadherin localization observed in 2D substrates could be different in 3D collagen gels. We have now performed immunofluorescence of E-cadherin in 3D-collagen embedded cells (**Fig.S4a**). We did not observe an obvious change in E-cadherin recruitment at cell-cell junctions in KO cells in those conditions. This supports now better our conclusion that E-cadherin is not involved in the decreased early cell-cell junction stability in KO cells.

We disagree with the reviewer that there is a clear difference in E-Cadherin recruitment in WT vs KO and KI cells. Indeed, quantification of E-cadherin recruitment at AJs does not show any difference for the 3 cell lines (**Fig.R2**). Nevertheless, E-cadherin recruitment after 6h in a confluent monolayer, was not expected to be similar to immediate E-cadherin recruitment of sparse cells establishing a junction.

It is important to point out that in cell monolayers, the only difference observed 6h after cell seeding is the recruitment of Arp2/3. Arp2/3 at AJs is decreased in KO monolayers, and increased in KI monolayers. This correlates closely with cell-cell junction stability phenotypes (also refer to our answer of the next question).

Figure removed per authors' request by LSA Editorial Staff.

6. The authors conclude that the vinculin-Arp2/3 interaction regulates cell-cell junction plasticity through Arp2/3 recruitment (e.g. p4). It is unclear on which data this is based, because the lack of junctional Arp2/3 recruitment in P878A mutant cells (fig. 6) does not lead to any observed changes in junctional phenotype.

We respectfully disagree with this comment of the reviewer, because the lack of junctional Arp2/3 recruitment in KI P878A cell line, indeed, do lead to a change in cell-cell junction stability, which is increased. We distinguish in our manuscript cell-cell junction stability versus plasticity. The latter refers, in WT cells, to the weakening of highly stable junctions in KI cells by the vinculin-Arp2/3 interaction. This plasticity is not dependent on E-cadherin recruitment at junctions (refer to our previous answer). The revised text has been modified in the Result and in the Discussion sections to clarify our interpretation of the data.

7. In Fig 1a, the authors should show the levels of ARPC1B and ARPC3 in the total lysates in all conditions.

As the reviewer wants to know whether Arp2/3 levels is modified upon linker expression, we now provide expression levels of the ARPC3 subunit in **Fig.1**. We did not observe any change in the levels Arp2/3 expression upon linker expression of the or the linker P878A.

8. The authors conclude that their data demonstrate vinculin's interaction with the canonical Arp2/3 complex in MCF10A cells. I recommend nuancing this conclusion, as the interaction is only demonstrated with the truncated linker and not full-length vinculin. Furthermore, enrichment of these canonical Arp2/3 proteins at cell-cell junctions (Suppl. Fig 5) does not demonstrate an interaction with vinculin.

We agree and thank the reviewer for this comment. We did not intend to overinterpret our data. Our point was to report which subunits of the Arp2/3 complex interacts with vinculin through the Arp2/3 binding motif of vinculin. We modified the Result and Discussion sections to nuance this conclusion.

Reviewer #2 (Comments to the Authors (Required)):

This is an interesting study focusing on the cellular roles of vinculin - Arp2/3 complex interactions. By careful analysis of wild-type, vinculin knockout, and vinculin knock-in cells (with a mutation in the Arp2/3-binding site), as well as by over-expressing wild-type and mutant versions of the vinculin 'linker region', the authors provide evidence that vinculin-Arp2/3 interactions inhibit actin polymerization and membrane protrusions, whereas the molecular clutch coupling branched actin to cell adhesions is not dependent on vinculin's interaction with the Arp2/3 complex. Moreover, they provide evidence that vinculin recruits Arp2/3 to cell-cell junctions and inhibits cell cycle in an Arp2/3-dependent manner.

The authors have used several cutting-edge approaches in these studies, and the data appear convincing and provide important new information on the interplay between vinculin and the Arp2/3 complex. However, there are few relatively minor points that should be addressed to further strengthen this manuscript.

1. The introduction to the vinculin - Arp2/3 interaction (on page 4 of the manuscript) was bit superficial, and the authors should more precisely explain here what is the evidence (based on the previous publications) for the direct interaction between vinculin and Arp2/3 complex, and what is already known/reported about the biochemical effects of vinculin on the Arp2/3 complex and Arp2/3-nucleated actin filament networks.

It is true that the Introduction on the vinculin-Arp2/3 interaction is concise. Indeed, little is known on the function of this interaction. To our knowledge, only 3 studies reported the vinculin-Arp2/3 interaction (DeMali et al., 2002, Moese et al., 2007; Chorev et al., 2014). The introduction section has been modified on page 4 to include evidences in favor of a direct interaction, from the two studies of DeMali and colleagues in 2002, and Chorev and colleagues in 2014. The first study was able to reconstitute the complex in vitro (Demali et al., 2002). The mass spectrometry analysis of purified vinculin-Arp2/3 complexes did not show the presence of additional protein (Chorev et al., 2014). This also suggests that the interaction could be direct in those extracts. However, the biochemical effects of vinculin on Arp2/3 remains largely unexplored (Romero et al 2020 for review).

2. Similarly, the authors should at least speculate in the 'Discussion' what are the possible molecular mechanisms by which vinculin affects the localization and /or activity of the Arp2/3 complex. Because based on Fig. S2 vinculin does not affect VCA-induced actin filament assembly by the Arp2/3 complex, is it possible that vinculin would affect the stability of Arp2/3-nucleated branches through GMF or cortactin? Alternatively, vinculin could sequester Arp2/3 complex from the membrane-associated NPFs to inhibit Arp2/3-catalyzed actin filament assembly in cells. To examine the latter option, the authors could just simply analyze the localization of the Arp2/3 complex in their wild-type, vinculin knockout, and vinculin knock-in cells (to see if e.g. in the knock-in cells, the intensity of Arp2/3 would be decreased at the edges of lamellipodia).

We agree with the reviewer. This is a logical suggestion, which is shared by **reviewer #3**. It is true that we did not want to speculate too much in the initial manuscript. We have now detailed possibilities of vinculin inhibiting branched actin assembly in the last paragraph of the Discussion section.

We investigated (also suggested by **reviewers #1**) the localization of Arp2/3 at the leading edge of lamellipodia (revised **Fig.3**). The main difference between parental vs KO and KI cell lines is that Arp2/3 is recruited slightly deeper in the lamellipodium in KO and KI cells. Thus, the vinculin-Arp2/3 interaction decreases the overall amount of Arp2/3 in the lamellipodium, which supports our original observation of lamellipodial properties of KO and KI cells. Unfortunately, this does not tell much more about the mechanism of inhibition of branched actin assembly by the vinculin-Arp2/3 interaction. Indeed, the decreased amount of lamellipodial Arp2/3 by the vinculin-Arp2/3 interaction can result either from vinculin inhibiting or sequestering Arp2/3, or vinculin destabilizing branches to accelerate branched actin depolymerization at the back of the lamellipodium. Nevertheless, we thank the reviewer for this suggestion, which was valuable for the quality of our manuscript.

3. The image quality was not particular good (especially in Figs. 1, and 2), and thus the authors may consider providing better quality immunofluorescence images and time-lapse images from the movies.

We agree with the reviewer that fluorescence microscopy images in **Fig.1-3** was not good enough. We realized that this was due to a loss of quality when downsizing original images. Indeed, original images at full size, acquired on cutting edge microscopes (Leica SP8-X confocal and Deltavision OMX SR TIRF-SIM), were of high quality. We have now worked on reorganizing those figures 1-4 to keep a larger amount of space for the TIRF-SIM and confocal images.

4. The analysis of protrusion speed/rearward flow/actin assembly rate/protrusion efficiency was bit confusing. It appears that the authors have just analyzed protrusion speed and actin rearward flow from the obtained kymographs, and thus plotting also the actin assembly rate and protrusion efficiency in the figures is not particularly informative.

We apologize that this section of the text confused the reviewer. The reviewer understood well our approach, which is to measure protrusion speed and the retrograde flow rate directly in kymographs. To analyze actin dynamics, the classical way is to calculate the actin assembly rate and the protrusion efficiency from the latter values (Fäßler et al., 2023). By analyzing the 4 properties of protrusions, we were able to draw our conclusions: the vinculin-Arp2/3 interaction decreases branched actin assembly, and is not involved in the molecular clutch. Without those calculations, the logical expectation would be that the actin assembly rate in KO should differ from that of KI cells. This is not the case, and the increased values of actin assembly in KO and KI cells only result from the lack of the vinculin-Arp2/3 interaction. Protrusion efficiency, increased only in KI cells, results from the enhanced actin assembly and the unchanged molecular clutch.

Thus, those data are still present in the revised manuscript, but we have reworded the Results section for an easier understanding.

5. What are the 'control cells' in Fig. 2D & E, Fig 3E, and Fig. 4 B & G? How do these differ from the wild-type MCF10A cells? This should be specified.

We apologize for omitting the description of control cells in the legends of **Figures 2, 3 and 4**. Ctrl cells refers to genome edited cells, in which a control gRNA was used instead of a gRNA targeting vinculin. This has been added to the legends of corresponding figures.

6. The sentence: 'We found that antibiotic-selected MCF10A cells down-regulated the exogenous expression of tagged full-length vinculin in an increasing number of cells over time, but not when the construct was limited to the vinculin linker that connects the head to the tail and which contains the Arp2/3 binding site.' in the beginning of 'Results' is difficult to follow, and some additional information or explanation is needed here for clarification.

We agree that the sentence could have been more explicit. When we selected clones of cells stably transfected with WT-vinculin or P878A-vinculin, either in the parental or KO cell line, we unexpectedly observed that vinculin expression rapidly decreased over time, as shown in Fig.R2. This was not the case for parental MCF10A cells that were stably transfected with the vinculin linker construct and used in **Fig.1**. In those cells, expression level of the vinculin linker remained constant over time, as expected for a stable cell line. The revised text has been modified to clarify this aspect.

Figure removed per authors' request by LSA Editorial Staff.

7. Scale bar is missing from Fig. S6.

We apologize for this mistake thank the reviewer for spotting the absence of the scale bar in the figure. The figure has been modified and the scale bar is now present in **Fig.S7** (former **Fig.S6**).

Reviewer #3 (Comments to the Authors (Required)):

In this manuscript, James et al. investigate the interaction between vinculin and the Arp2/3 complex, and how this interaction regulates cell migration and proliferation. Contrary to previous studies, the authors find that the vinculin-Arp2/3 interaction inhibits the activity of Arp2/3. While the mechanism of inhibition by the vinculin linker remains largely unexplored, the presented data supports this notion. The authors might consider speculating on the potential mechanisms through which the vinculin linker inhibits Arp2/3 activity to further strengthen their manuscript. The brief mention of post-translational modification invites further speculation which could enrich the discussion.

Additionally, the use of epithelial cells instead of fibroblasts to demonstrate the cellular-context-dependent role of vinculin is a key and novel finding of this study. The authors could elaborate on the specific differences between epithelial cells and fibroblasts that account for their observed behaviors.

We agree with the reviewer, who shares the point of view of **reviewer #2** on the mechanistic aspect of branched actin inhibition. As said in the response to **reviewer#2** (**see point #2.**), it is true that we did not want to speculate too much on how vinculin inhibits branched actin assembly in the initial manuscript, in order to avoid overinterpretation of our data. We believe that branched actin inhibition by the vinculin-Arp2/3 could result either from the direct inhibition of Arp2/3, or the destabilization of branched actin networks. The revised version of the text has been modified as requested in the last paragraph of the Discussion section.

We agree with the reviewer that it is important for the community that we report the specificity of vinculin role in MCF10A cells. Indeed, the specific program of epithelial cells, required to build monolayers for example, differs from the one of fibroblasts that do not harbor similar cell-cell junctions. Similarly to specific effects of vinculin in epithelial cells, we reported a cell-type specific behavior induced by vimentin expression, which decreases migration persistence of MCF10A cells but has the opposite effects in most other cell types (Fokin et al, 2024). We have slightly modified the Discussion section to incorporate these observations.

Minor concerns below need to be addressed or clarified for the final version of the manuscript:

1. Data for knock-out (KO) and knock-in (KI) are often presented separately, making it challenging to compare and analyze the data holistically. It may be beneficial to combine the data into a single graph. Additionally, the term "KI" should be clarified, especially in the figures; for instance, using "KI-P878A" instead of "KI" to denote vinculin P878A knock-in.

We agree with the reviewer that the combination of the data in a single figure would facilitate the direct comparison of KO and KI cell lines. Unfortunately, in those cases, the experiments were performed separately, preventing us to combine KO and KI data on a single graph. Indeed, the KI cell line, which was very difficult to generate, was obtained a long time after the characterization of the two KO cell lines. Thus, data of KO and KI could not be combined in the figures, although phenotypes of KO and KI-P878A cell lines are always compared to that of parental cells.

As requested by the reviewer, the revised text has been modified and we now use KI-P878A instead of KI to refer to vinculin P878A knock-in.

2. On page 7, the statement that "Arp2/3 binding is not required for the focal adhesion (FA)-related functions of vinculin" is ambiguous. The localization and length of FAs may not sufficiently conclude their adhesive function. Including assays for adhesion strength to the extracellular matrix may help clarify the functional roles of this interaction.

We thank the reviewer for this valuable suggestion. We agree that our data did not provide a definite proof that the vinculin-Arp2/3 interaction is not involved in FA-related functions. However, at the single cell level, the aim of our report was to investigate the role of vinculin on branched actin functions. Our motivation was fueled by the analysis of cell migration, in which persistence was dramatically increased in KO and KI cells, whereas other parameters of cell migration, such as retraction of the cell body or FA morphology were not obviously changed. As suggested by the reviewer, we have included in the revised version of the manuscript adhesion assays to clarify the function of the vinculin-Arp2/3 interaction on cell adhesion to the ECM (**Fig.S3b**). We did not observe significant changes in adhesion of KO and KI cell lines, supporting our initial conclusion.

3. On page 8, the authors state that cell-cell interactions were "not obviously affected in KO cells" on 2D surfaces. However, Supplementary Movie 7 shows more single cells at the leading wound edge in KO cells, which aligns with their 3D data. Discussing this observation could support their model.

This is a very interesting comment pinpointed by the reviewer. We fully agree with him that the fact that more single KO cells are observed at the front edge of the wound supports the decreased cell-cell junction stability observed in 3D collagen matrix. We added this observation in the Results and Discussion sections of the revised manuscript.

4. The phrase "the mechanotransducer function of vinculin" on page 9, and the deduction of vinculin's mechanosensitive function from their data, are unclear. It may be more accurate to suggest that the vinculin-Arp2/3-independent function is necessary for the stability of cell-cell junctions.

To date, the main described function of vinculin involves the reinforcement of cell adhesions in response to mechanical forces, either from extracellular mechanical cues or actomyosin-based intracellular forces applied on the talin/alpha-catenin module (Atherton et al., 2016; Ladoux et al., 2015 for reviews). However, we agree with the reviewer that we did not address the mechanosensitive function of vinculin in our manuscript. As suggested by the reviewer, we slightly modified our conclusion on the Arp2/3-independent functions of vinculin in the Discussion section.

5. The sentence on page 9, "In contrast, the interaction of vinculin with Arp2/3 is not required for the stability of cell-cell junctions, but rather enhances cell-cell adhesions," is confusing. Given the complexity of 3D cell migration, vinculin might play multiple roles, such as in 3D cell-ECM and cell-cell interactions, which could differ from their 2D counterparts. The evidence based on dissociation events might not sufficiently demonstrate the strength of cell-cell adhesion. The authors could clarify or reword this sentence to avoid confusion.

We apologize that our sentence was misinterpreted. Indeed, we did not want to overinterpret our results in terms of cell-cell adhesion strength. Our point was to conclude that cell-cell junctions were less stable in the absence of vinculin (KO cells). Thus, this indicates that the vinculin-Arp2/3 interaction is not required to establish stable cell-cell adhesion. Because these junctions are slightly more stable when vinculin do not interact with Arp2/3, this suggests that the interaction could play a role in the remodeling/plasticity of cell-cell junctions. The revised text has been reworded to clarify our conclusion.

References

Atherton P, Stutchbury B, Jethwa D & Ballestrem C (2016) Mechanosensitive components of integrin adhesions: Role of vinculin. *Exp Cell Res* 343: 21–27

Chorev DS, Moscovitz O, Geiger B & Sharon M (2014) Regulation of focal adhesion formation by a vinculin-Arp2/3 hybrid complex. *Nature communications* 5: 1–11

DeMali KA, Barlow CA & Burridge K (2002) Recruitment of the Arp2/3 complex to vinculin: coupling membrane protrusion to matrix adhesion. *The Journal of Cell Biology* 159: 881–891

Fäßler, F., M.G. Javoor, J. Datler, H. Döring, F.W. Hofer, G. Dimchev, V.-V. Hodirna, J. Faix, K. Rottner, and F.K.M. Schur. 2023. ArpC5 isoforms regulate Arp2/3 complex-dependent protrusion through differential Ena/VASP positioning. *Sci Adv*.

Fokin, A.I., A. Boutillon, J. James, L. Courtois, S. Vacher, G. Simanov, Y. Wang, A. Poleskaya, I. Bièche, N.B. David, and A.M. Gautreau. 2024. Inactivating negative regulators of cortical branched actin enhances persistence of single cell migration. *J Cell Sci*. 137:jcs261332. doi:10.1242/jcs.261332.

Ladoux, B., W.J. Nelson, J. Yan, and R.M. Mège. 2015. The mechanotransduction machinery at work at adherens junctions. *Integr Biol (Camb)*. 7:1109–1119. doi:10.1039/c5ib00070j.

Moese S, Selbach M, Brinkmann V, Karlas A, Haimovich B, Backert S & Meyer TF (2007) The Helicobacter pylori CagA protein disrupts matrix adhesion of gastric epithelial cells by dephosphorylation of vinculin. *Cell Microbiol* 9: 1148–1161

Romero, S., C.L. Clainche, and A.M. Gautreau. 2020. Actin polymerization downstream of integrins: signaling pathways and mechanotransduction. *Biochemical Journal*. 477:1–21. doi:10.1042/bcj20170719.

October 25, 2024

RE: Life Science Alliance Manuscript #LSA-2024-02583-TR

Dr. Stéphane Romero
École Polytechnique
BIOC, CNRS UMR7654
Route de Saclay
Palaiseau 91128
France

Dear Dr. Romero,

Thank you for submitting your revised manuscript entitled "Vinculin-Arp2/3 Interaction Inhibits Branched Actin Assembly to Control Migration and Proliferation". We would be happy to publish your paper in Life Science Alliance pending final revisions necessary to meet our formatting guidelines.

- please be sure that the authorship listing and order is correct
- please add the Twitter handle of your host institute/organization as well as your own or/and one of the authors in our system
- you may want to consider uploading Figure 9 as a Graphical Abstract rather than as a figure, but this is up to you

Figure Check:

- please add sizes next to all blots

A. FINAL FILES:

B. MANUSCRIPT ORGANIZATION AND FORMATTING:

Thank you for your attention to these final processing requirements. Please revise and format the manuscript and upload materials within 5 days.

Sincerely,

Reviewer #2 (Comments to the Authors (Required)):

The authors have satisfactorily addressed my previous comments.

Reviewer #3 (Comments to the Authors (Required)):

We appreciate the authors' thorough response and revision of the manuscript, which have satisfactorily resolved all concerns from our perspective.

November 5, 2024

RE: Life Science Alliance Manuscript #LSA-2024-02583-TRR

Dr. Stéphane Romero
École Polytechnique
BIOC, CNRS UMR7654
Route de Saclay
Palaiseau 91128
France

Dear Dr. Romero,

Thank you for submitting your Research Article entitled "Vinculin-Arp2/3 Interaction Inhibits Branched Actin Assembly to Control Migration and Proliferation". It is a pleasure to let you know that your manuscript is now accepted for publication in Life Science Alliance. Congratulations on this interesting work.

DISTRIBUTION OF MATERIALS:

Again, congratulations on a very nice paper. I hope you found the review process to be constructive and are pleased with how the manuscript was handled editorially. We look forward to future exciting submissions from your lab.

Sincerely,
